# Sirt6 ablation in the liver causes fatty liver that increases cancer risk by upregulating Serpina12

Licen Li[1], Jianming Zeng[1], Xin Zhang [1], Yangyang Feng[1], Josh Haipeng Lei [1], Xiaoling Xu[1,2], Qiang Chen [1,2✉] & Chu-Xia Deng [1,2✉]

## Abstract

**Non-alcoholic fatty liver disease is a chronic liver abnormality that exhibits high variability and can lead to liver cancer in advanced stages. Hepatic ablation of SIRT6 results in fatty liver disease, yet the potential mechanism of SIRT6 deficiency, particularly in relation to downstream mediators for NAFLD, remains elusive. Here we identify *Serpina12* as a key gene regulated by *Sirt6* that plays a crucial function in energy homeostasis. Specifically, *Sirt6* suppresses *Serpina12* expression through histone deacetylation at its promoter region, after which the transcription factor, Cebpα, binds to and regulates its expression. Sirt6 deficiency results in an increased expression of Serpina12 in hepatocytes, which enhances insulin signaling and promotes lipid accumulation. Importantly, CRISPR-Cas9 mediated *Serpina12* knockout in the liver ameliorated fatty liver disease caused by Sirt6 ablation. Finally, we demonstrate that Sirt6 functions as a tumor suppressor in the liver, and consequently, deletion of Sirt6 in the liver leads to not only the spontaneous development of tumors but also enhanced tumorigenesis in response to DEN treatment or under conditions of obesity.**

**Keywords** Fatty Liver; Lipid Homeostasis; Liver Cancer; Serpina12; Sirt6
**Subject Category** Metabolism

## Introduction

Non-alcoholic fatty liver disease (NAFLD) is a condition characterized by fat deposition in liver cells without consuming alcohol or exposure to hepatotoxic substances (Cohen et al, 2011; Powell et al, 2021). Despite its prevalence at around 25%, NAFLD continues to increase worldwide, making it a serious health threat(Yki-Jarvinen, 2016). The liver is the energy metabolic center responsible for maintaining the lipid balance to protect the body from lipotoxicity (Mashek, 2021; Trauner et al, 2010). Once it is dysregulated, lipids would accumulate and result in the formation of fatty liver. This chronic condition is usually overlooked initially thus leading to progress to liver cancer in the late stage (Cohen et al, 2011; Loomba et al, 2021; Marengo et al, 2016). Mechanisms of the progression of liver disease are still not fully understood. Notably, fatty liver disease and liver cancer share similar features of metabolic reprogramming (Cheng et al, 2018; Nakagawa et al, 2018; Ward and Thompson, 2012). In liver cancer, malignant cells require more lipids to support their proliferation and growth (Currie et al, 2013; Liu et al, 2017; Santos and Schulze, 2012). In HFD-driven hepatocellular carcinoma (HCC), fatty acid oxidation (FAO) is decreased leading to acylcarnitine accumulation, which promotes carcinogenesis (Fujiwara et al, 2018; Sangineto et al, 2020). Thus, metabolic reprogramming especially lipid switch for cancer cells, requires modulators to coordinate this process to ensure its proliferation.

Sirtuin 6 (SIRT6) belongs to the sirtuin family, which is the NAD+-dependent histone deacetylase (Finkel et al, 2009; Liszt et al, 2005; Seto and Yoshida, 2014). It has been involved in multiple processes including metabolic disease, aging as well as cancer (Chang et al, 2020). Both histone H3 lysine 9 (H3K9) and histone H3 lysine 56 (H3K56) are well-known targets of SIRT6 deacetylation (Michishita et al, 2008, 2009). To study the functions of Sirt6 in vivo, we have previously knocked out Sirt6 and found the Sirt6 deficiency generated by whole-body disruption resulted in post-weaning lethality in the majority of mice (Mostoslavsky et al, 2006; Xiao et al, 2010). We also found that feeding the Sirt6[-/-] mice with water containing 10% glucose could suppress the post-weaning lethality in about 80% of mutant mice. Up to 60% of the survivors also developed chronic liver inflammation at about 1 year of age (Mostoslavsky et al, 2006). Several Sirt6[−/−] mice developed fibrosis and cancer in the liver (Chen et al, 2018; Xiao et al, 2012). Of note, the mouse strain carrying liver-specific knockout of Sirt6 (Sirt6[Co/Co] Alb-Cre, or Sirt6-LKO) did not develop inflammation in the liver, suggesting that the chronic liver inflammation phenotype observed in the Sirt6 whole-body mutant mice might be caused by Sirt6 deficiency in other organs/tissues (Kim et al, 2010). The Sirt6-LKO mice were morphologically normal, and their blood glucose levels were comparable to those of control mice at the young stage (<8 months of age). However, it was revealed that their blood glucose levels were slightly increased after glucose and insulin challenge conditions (≥8 months of age) (Kim et al, 2010). Disruption of Sirt6 in the liver resulted in the gradual accumulation of lipids, which became obvious in older stages (from 7.5 to 13 months), with almost all mutant mice (90%) exhibiting fatty liver compared to only 12% of control mice

[1]Cancer Centre, Faculty of Health Sciences, University of Macau, Macau SAR, China. [2]MOE Frontier Science Centre for Precision Oncology, University of Macau, Taipa, Macau SAR 999078, China. ✉E-mail: qiangch@um.edu.mo; cxdeng@um.edu.mo

(Kim et al, 2010). In addition, the data also revealed that fatty liver formation was accompanied by numerous changes in gene expression leading to enhanced glycolysis and decreased β-oxidation (Kim et al, 2010). Although this study demonstrated the critical role of Sirt6 in maintaining metabolic homeostasis for optimal liver function, the downstream mediators that mediate Sirt6 function remain elusive. Furthermore, since Sirt6-LKO mice were studied for no longer than 13 months and no tumors were observed in this stage, the contribution of Sirt6 to liver cancer development remains unknown (Kim et al, 2010). Fatty liver diseases have been implicated in liver cancer formation (Marengo et al, 2016), therefore, we would like to investigate whether Sirt6 deficiency could trigger the onset of liver cancer in advanced stages and elucidate the underlying mechanism for cancer formation.

In the present study, we have identified a novel target, *Serpina12*, that is suppressed by Sirt6 in the liver and contributing to the onset of liver steatosis when Sirt6 is deficient. Notably, we have found that Sirt6-LKO mice exhibit the spontaneous development of liver tumors at approximately two years of age, while also being susceptible to tumor formation induced by diethylnitrosamine (DEN) treatment or *ob/ob* related obesity. This study provides clear and direct evidence that Sirt6 acts as a tumor suppressor in the liver.

## Results

### Sirt6 ablation results in fatty liver with alteration of a broad lipid-related gene expression and an increase in H3K9 and H3K56 acetylation

Because our earlier work on Sirt6-LKO mice did not observe liver cancer formation up to 13 months of age although they all developed fatty liver (Kim et al, 2010), we first divided Sirt6-LKO mice into several cohorts for the development of spontaneous liver cancer in aging mice; chemically induced tumorigenesis after treating with DEN; and obesity-induced tumorigenesis by crossing with ob/ob mutant mice (Fig. 1A). Meanwhile, to identify direct downstream mediators of Sirt6 in connection with Sirt6 deficiency and fatty liver, we performed quantitative transcriptome analysis using RNA sequencing (RNA-seq) from the livers of Sirt6$^{co/co}$ (Sirt6 Floxed) and Sirt6-LKO mice at 8 months old age, when the mutant mice suffered from fatty liver (Fig. EV1A) due to the targeted disruption of Sirt6, as reflected by markedly reduced mRNA and protein levels (Fig. EV1B,C). The RNA-seq identified 227 upregulated and 212 downregulated differentially expressed genes (DEGs) in 8-month-old Sirt6-LKO mice compared to control mice (Fig. 1B). For the upregulated DEGs, most of them were enriched in the lipid metabolic process (Fig. 1C). While the downregulated DEGs were enriched in the immune response process (Fig. EV1D). To identify the direct target of Sirt6, we focused on upregulated genes because Sirt6-deficiency is known to cause chromatin opening leading to an increase in gene expression. The most marked upregulated DEGs were enriched in arachidonic acid metabolism and biosynthesis of unsaturated fatty acids in GSEA analysis (Fig. 1D), indicating dysregulation of lipid homeostasis in the liver after Sirt6 ablation.

Sirt6 represses transcription by acting as a specific deacetylase for H3K9ac and H3K56ac (Michishita et al, 2008, 2009). To identify

the genome regions targeted by Sirt6-dependent histone deacetylation impacting gene expression, we performed a chromatin immunoprecipitation sequencing (ChIP-seq) analysis in Sirt6 Floxed and Sirt6-LKO mice using antibodies targeting H3K9ac and H3K56ac, respectively. The enhanced binding peaks of H3K9ac and H3K56ac in the liver occur most frequently near the transcription start site (TSS) of gene bodies based on metagene analysis within a 10 kb range (Fig. EV1E). We then mapped the binding in a genome-wide manner and revealed a stronger binding pattern in the TSS at 2 kb region of gene bodies on Sirt6 LKO than Sirt6 Floxed liver. 2816 and 6127 gained sites were shown in H3K9Ac and H3K56Ac respectively in Sirt6 LKO mice (Fig. 1E), suggesting these genes were mostly affected by Sirt6-dependent histone marks. Next, we compared the RNA-seq upregulated DEGs and genes whose TSS have increased binding of H3K9ac or H3K56ac, and identified 93 and 47 genes respectively (Fig. EV1F), indicating that these genes were the most promising Sirt6-dependent histone deacetylation target genes. Of note, these 47 genes in the H3K56ac upregulation group were completely overlapped in H3K9ac with RNA-seq upregulated DEGs (Fig. EV1F), and the increased expression pattern in Sirt6 LKO was confirmed by RNA-seq (Figs. 1F and EV1G). Thus, we identified a small number of candidate genes that were the most promising targets for Sirt6 associated with fatty liver development in Sirt6 LKO mice (Fig. EV1G).

### Sirt6 ablation upregulates Serpina12 mediated by transcription factor CEBPα

Our RT-PCR analysis revealed that 29/47 genes were significantly upregulated in the Sirt6 mutant liver tissue (Fig. EV2A). We believed that the direct downstream target genes of Sirt6 deacetylase should contain H3K9ac and H3K56ac binding sites in their promoter region, in addition to being upregulation in the 3- and 8-month Sirt6 mutant livers. Based on this criterion, five genes were identified (Figs. 2A,B and EV2B). Among them, *Serpina12*, *Cyp2b10*, *Gal3st1*, and *Orm3* were found to be significantly upregulated after further validation in the liver tissue (Fig. 2C). Therefore, we obtained 4 candidate genes that meet the specified criteria. The Sirt6 LKO mice we used are based on our previous work using the *Cre/loxP* system. The Cre recombinase is under the control of a mouse albumin enhancer/promoter whose activity is specific to hepatocytes (Kim et al, 2010). We then isolated the primary hepatocytes and performed an acute knockout of Sirt6 in vitro to further validate these candidate genes. Of note, *Serpina12* and *Cyp2B10* were significantly increased in the primary hepatocytes (Fig. 2D). Cyp2b10 belongs to the cytochrome p450 (CYP) family, and its activation has been a direct target of the constitutive androstane receptor (CAR), which is well-known as a liver tumor promoter that has been demonstrated previously (Kettner et al, 2016). Serpina12 is also shown to promote HCC through the hyperactivation of AKT/β-catenin signaling (Yu et al, 2023). In this study, we focus on the most significantly changed gene, *Serpina12*, as our candidate for further analysis. Our results showed that once Sirt6 was lost, Serpina12 was highly expressed in liver tissue at different ages in mice at both mRNA (Fig. 2E), and protein levels (Fig. 2F) in the Sirt6 LKO mice. We next performed acute knockdown of SIRT6 in the HepG2 cells with different shRNAs and found that SERPINA12 expression increased after

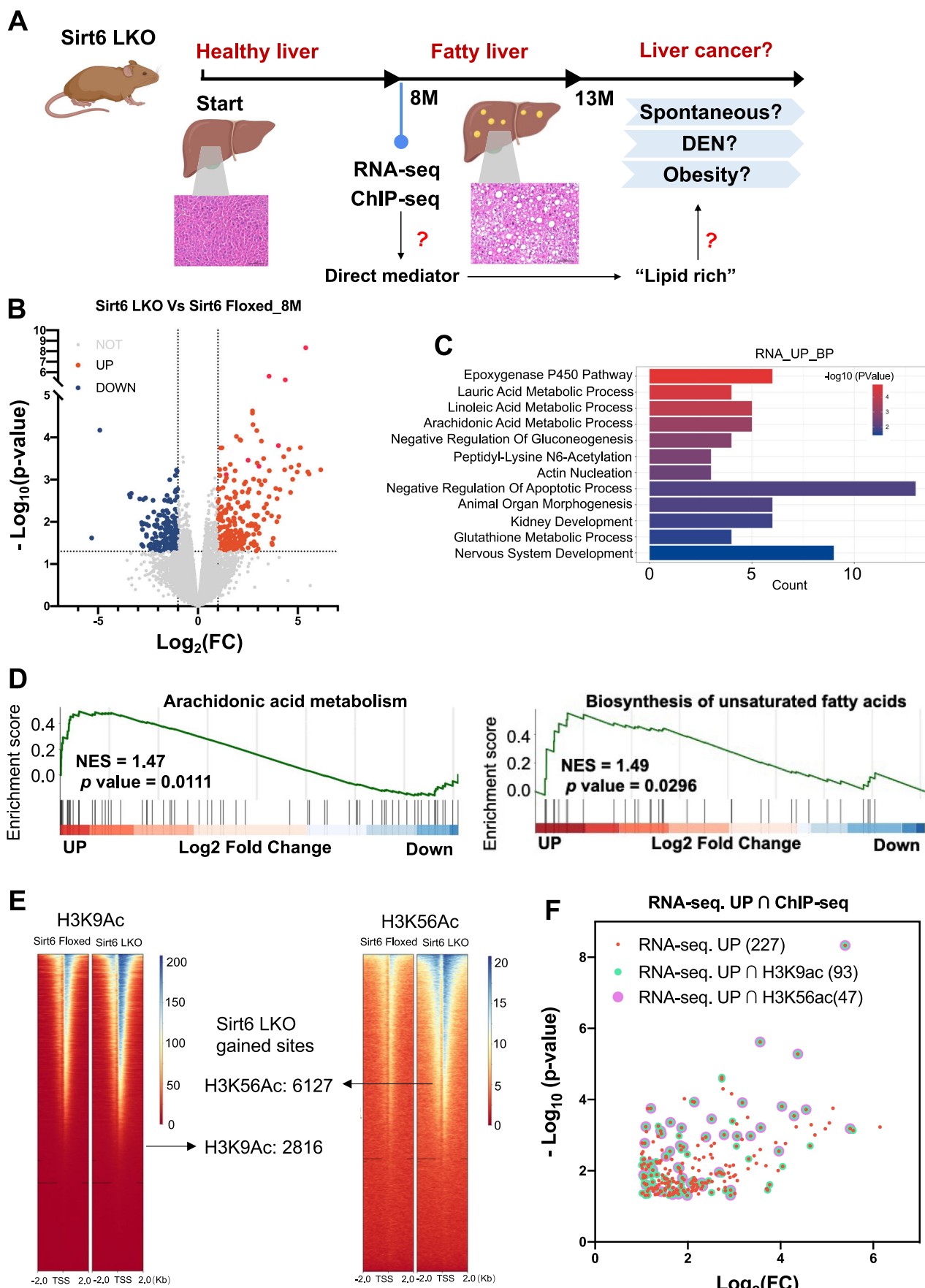

**Figure 1.   Sirt6 ablation induces formation of fatty liver accompanied by alterations of a broad lipid-related gene expression and increased H3K9 and H3K56 acetylation.**

(A) The schematic model illustrates experimental design and questions to be investigated using the Sirt6 LKO mice. This panel was reused in the paper. (B) Volcano plots showing the 13659 expressed protein coding genes (RPM > 1). Among these genes, 227 genes are upregulated, and 212 genes are downregulated at significant level. (C) RNA-seq upregulated different express genes enriched based on the biological process. (D) GSEA showing the upregulated genes enriched in arachidonic acid metabolism and biosynthesis of unsaturated fatty acids. (E) ChIP-Seq showing the enriched peaks in Sirt6 Floxed or Sirt6 LKO mice liver with either antibody against H3K9ac or H3K56ac. (F) Scatter plot showing that RNA-sequence upregulated genes with overlapped ChIP-H3K9Ac/H3K56Ac peaks. Data information: For (B),(C),(D), and (F), $n = 3$ biological replicates (BR), *t*-test, *p* value < 0.05, Log$_2$ FC > 1 or < −1. Source data are available online for this figure.

SIRT6 knockdown (Fig. 2G,H). Altogether, our data indicate that SIRT6 negatively regulates SERPINA12 and SIRT6 deficiency consequently results in its upregulation.

Next, we checked the levels of H3K9ac and H3K56ac in the genome region of Serpina12 based on ChIP-seq analysis. The Integrative Genomics Viewer (IGV) showed that both regions in the Sirt6-deficient group near the TSS sites contained more binding peaks (Fig. 3A). We then validated this enriched binding peak with ChIP-qPCR. Indeed, there were significant enhancements in H3K9ac and H3K56ac levels in the Sirt6-LKO liver compared to the control (Fig. 3B). To determine which transcription factors might regulate Serpina12 expression, we analyzed ChIP-seq data and found that the active binding region was located near the TSS of Serpina12 in the Sirt6 deficient group (Fig. 3C). Furthermore, we analyzed this region using online software (Genomatix Software Suite) and found several potential transcription factors that may bind to it, including Cebpα, Pparγ, and Srebf1 (Fig. 3C). To verify whether these transcription factors regulate the expression of Serpina12, we co-transfected the promoter of Serpina12 with these transcription factors into cell lines and analyzed it using a Luciferase Reporter assay. It was found that CEBPα significantly enhanced the activity of luciferase compared to the control group (Figs. 3D and EV3A,B). Indeed, we found nearly 90% similarity consensus for CEBPRE located near the TSS of the *Serpina12* locus (Aibara et al, 2019) (Fig. EV3C). CEBPα is a CCAAT/enhancer binding protein, which is a basic leucine zipper transcription factor. It is critical for regulating glucose and lipid metabolism. To determine whether Sirt6 regulates the expression of Serpina12 through CEBPα, we knocked out Sirt6 in primary hepatocytes and co-transfected the CEBPα and Serpina12 promoter into these cells. It was found that Sirt6 deletion enhanced the luciferase activity significantly (Fig. 3E), suggesting that Sirt6 affects the transcription factor CEBPα to regulate the expression of Serpina12. Besides, we performed ChIP by using the antibody targeting CEBPα, and the qPCR results showed that an increased in binding signal in Sirt6 KO primary hepatocytes (Fig. 3F). To determine whether C/EBPa could directly interact with Sirt6, we induced Sirt6-GFP and C/ebpα expression 48 h in 293T cells and performed a co-immunoprecipitation assay with antibodies against Sirt6 and C/ebpα. The results showed that C/ebpα could directly interact with Sirt6 (Fig. EV3D). The binding of C/ebpα is important for Sirt6 to suppress the expression of Serpina12 as such suppression was largely abolished in the mutant Serpina12 luciferase promoter activity that lacks functional C/EBPa binding site (Fig. EV3E). To examine the affects in human cell line, we first identified the SERPINA12 promoter region and observed that region 3 should be affected by SIRT6 (Fig. EV3F). Next, after knocking down SIRT6 in HepG2 cells (Fig. EV3G), SERPINA12 showed more enhanced binding when using antibody target H3K9ac (Fig. 3G) and CEBPα

(Fig. 3H). If we double knocked down SIRT6 and CEBPα together, SERPINA12 expression was suppressed in HepG2 cells (Fig. EV3H). Therefore, SIRT6 primarily acts as an H3K9 deacetylase to regulate SERPINA12 expression. In addition, the transcription factor CEBPα could also bind to the SERPINA12 promoter to regulate its expression. Furthermore, knockdown of CEBPα overrode the effect of Sirt6 on Serpina12 expression (Fig. EV3H).

## Serpina12 enhances insulin signaling and promotes lipogenesis in the liver of Sirt6 deficient mice

As the liver is the center of metabolic regulation, we noticed that Serpina12 could only be detected in the liver after Sirt6 deficiency (Fig. EV4A). We then isolated primary hepatocytes from the Sirt6 Floxed mice liver to examine the potential role of Serpina12. Upon knockout or knockdown of Serpina12 (Figs. 4A and EV4B), we stimulated the cells with insulin and found that insulin signaling was impaired within the time (Figs. 4A and EV4C). When Serpina12 was overexpressed in the cells, the insulin signaling was increased as reflected by the level of insulin receptor substrate 1 (IRS1) phosphorylation level (Fig. 4B), suggesting it plays a role in metabolic homeostasis. Of note, knockout of Sirt6 under the same condition enhanced the insulin signaling (Fig. EV4D), suggesting that the loss of Sirt6 caused an increase in Serpina12 expression to mediate the insulin signaling. To examine this hypothesis, we infected primary hepatocytes with adeno-Cre virus to knock out Sirt6 and meanwhile knocked out or knocked down Serpina12, and found that insulin signaling also declined (Fig. 4A,C). Thus, the knockdown of Serpina12 led to a reversal of the enhanced insulin signaling caused by Sirt6 deficiency. As insulin levels are always accompanied by glucose alterations to maintain the blood glucose at a stable level, we thus used different doses of glucose to treat the hepatocytes and found that a high concentration of glucose could enhance the Serpina12 expression at both the mRNA and protein levels (Figs. 4D and EV4E). In addition, we observed that under high glucose conditions, Sirt6 KO generated more lipids in the cells, and the knockdown of Serpina12 could abolish the Sirt6 deficiency-caused lipogenesis (Figs. 4E and EV4F). The high glucose concentration resulted in increased triglyceride levels only in the Sirt6 KO-treated cells and dampened if knockdown Serpina12 was knocked down (Fig. 4E,F). To further confirm the role of Serpina12 in lipid metabolism, we established a lipogenesis model in HepG2 cells induced by oleic acid. Our data indicated that lipogenesis enhanced by Sirt6 deficiency was suppressed by knocking out SERPINA12 (Appendix Fig. S1A,B). Thus, SIRT6 deficiency enhanced insulin signaling is largely mediated by the upregulation of Serpina12 levels, and high glucose concentration could also enhance its expression to promote lipogenesis (Fig. 4G).

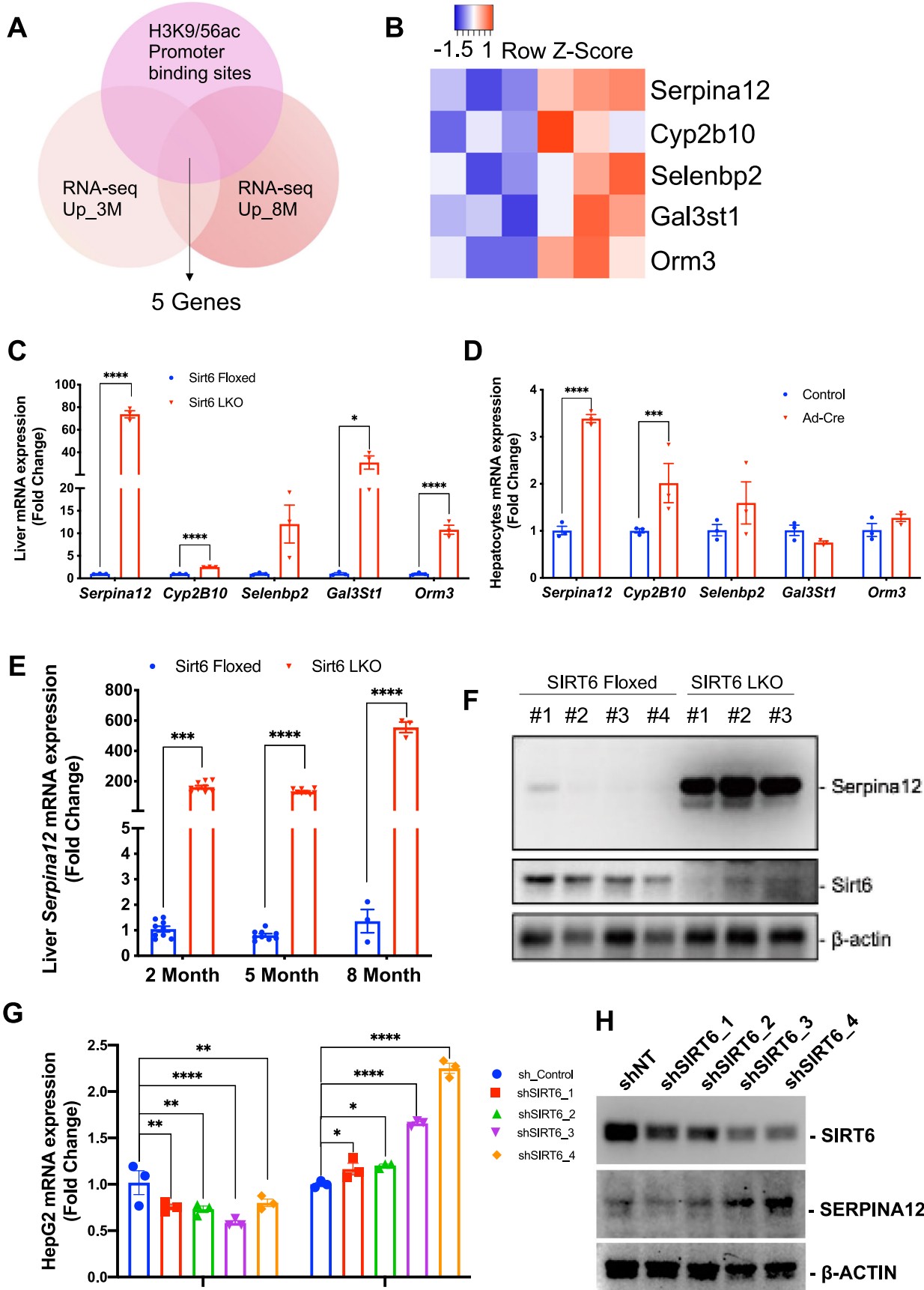

◄ **Figure 2. Serpina12 upregulated after Sirt6 ablation.**

(A) Venn diagram showing that the 5 candidate genes which overlapped with RNA-sequence upregulated genes in 3 months, 8 months and ChIP-H3K9Ac/H3K56Ac promoter binding enriched peaks. (B) Heatmap showed the expression level in RNA-sequence. (C) qPCR validation of the 5 candidate genes, $n = 3$ BR. (D) qPCR analysis of the candidate genes mRNA levels from either control or Ad-Cre in primary hepatocytes, $n = 3$ technical replicates (TR). (E) qPCR showing the Serpina12 mRNA level in 2 months, 5 months, and 8 months in Sirt6 Floxed or Sirt6 LKO mice liver. For 2 months and 5 months, $n = 9$ (3 BR x 3 TR); for 8 months, $n = 3$ TR. (F) Western blot shows the Serpina12 protein level in liver in Sirt6 Floxed or LKO mice. (G) qPCR showing the SIRT6 and SERPINA12 mRNA expression level with different SIRT6 shRNAs in HepG2 cells, $n = 3$ TR. (H) Western blot analysis of SIRT6 and SERPINA12 protein levels with different SIRT6 shRNAs in HepG2 cells. Data information: For (C),(D),(E), and (G), error bars = SEM. Multiple $t$-test or two-way ANOVA. *$p < 0.05$, **$p < 0.01$, ***$p < 0.005$, ****$p < 0.001$. Source data are available online for this figure.

## Effect of Serpina12 on the lipogenesis and fatty liver formation induced by Sirt6 deficiency

To understand how Serpina12 affects lipogenesis, we used primary hepatocytes with either knockout of Sirt6 or knockdown of Serpina12 at a high concentration of glucose. The data showed that the key enzymes for de novo synthesis of triglycerides (TGs) had increased, whereas they were decreased if Serpina12 was knocked down at a significant level (Fig. 5A). To examine the effect of Serpina12 in vivo, we established a Serpina12 knockout in Sirt6 LKO mice through hydrodynamic tail vein injection of CRISPR/Cas9 sgSerpina12 (Fig. 6B). To evaluate the sgSerpina12 cutting efficiency and specificity, we used a GFP reporter system, which could generate a functional GFP signal when sgSerpina12 works (Miao et al, 2019) (Appendix Fig. S2A). We designed seven sgRNA constructs and selected those with stronger GFP signals, and mixed for injection (Appendix Fig. S2A). After one year and four months, we harvested these mice and confirmed the knockout efficiency of Serpina12 in Sirt6 LKO mice (Appendix Fig. S2B,C). The liver sections showed a marked decrease in lipid droplets in the Sirt6-LKO with Serpina12 knockout group (Fig. 5B), suggesting that Serpina12 promotes lipogenesis in vivo. Besides, we checked that the glucose and lipid metabolism-related transcription factor, Pparg, had decreased in the double knockout group (Fig. 5C). Furthermore, the glucose metabolism key enzymes G6PC and LPK, as well as the fatty acid synthesis key enzymes fatty acid synthase (Fasn) and Elovl6, were decreased in the double knockout group at a significant level compared to Sirt6 LKO mice (Fig. 5C). Thus, Sperina12 promotes lipogenesis by affecting glucose and lipid-related gene expression (Fig. 5C). Furthermore, to understand the role of SERPINA12 in human NAFLD patients, we analyzed its expression in the GEO database (GSE89063). We observed that SERPINA12 expression was significantly increased in nonalcoholic steatohepatitis (NASH) patients compared with an in vitro human liver model (a condition of low glucose and low insulin to mimic the healthy liver) (Fig. 5D) (Feaver et al, 2016). The glucokinase (GCK) which promotes glucose metabolism was also increased in NASH as well as the lipid synthesis genes like the FASN and adenosine triphosphate (ATP) citrate lyase (ACLY) were increased (Fig. 5D). These data indicate that in NASH patients where lipogenic programs are increased, SERPINA12 is significantly increased.

We further explored The Cancer Genome Atlas (TCGA) data to check if SERPINA12 expression levels affect the overall survival of patients with HCC. We identified a correlation between the levels of SERPINA12, FASN, and patient survival. The FASN gene encodes an enzyme that catalyzes malonyl CoA to fatty acids to promote lipid synthesis and is frequently up-regulated in HCC (Che et al,

2017; Yamashita et al, 2009). We found that high SERPINA12 expression levels in combination with high FASN expression had the worst prognosis for overall survival (Fig. 5F) compared to the combination of high FASN expression with low SERPINA12 expression (Fig. 5E). Moreover, when we examined the histology of liver sections from these HCC patients, we observed that high SERPINA12 expression in combination with high FASN (Appendix Fig. S2D, bottom right) was associated with markedly different histology than that associated with low SERPINA12 expression and high FASN expression (Appendix Fig. S2D, top right). This indicates that high SERPINA12 in human HCC sections appeared to contain more lipid when associated with high FASN expression. Overall, these results indicate that high SERPINA12 expression is associated with increased lipid accumulation, more aggressive HCC, and worse patient outcomes when linked to high FASN expression.

## Sirt6 deficient causes lipid-rich environment to accelerate HCC formation in mice

Studying Sirt6 whole body knockout mice, we have previously shown that a small fraction of the mutant mice developed lesions and liver cancer triggered by inflammation (Xiao et al, 2012). Our examination of Sirt6-LKO mice at about two years of age clearly detected spontaneous HCC formation in 6 out of 7 (86%) mutant mice, which is more than triple as much as the control (Figs. 6A,B and EV5A), suggesting that Sirt6 is a tumor suppressor in liver cancer. To further investigate the hypothesis, we conducted two additional experiments. We first challenged Sirt6 Floxed and LKO mice with diethylnitrosamine (DEN), a proven liver carcinogen (Dapito et al, 2012; Tolba et al, 2015), on postnatal day 14 and examined cancer formation when mice reached 7 months of age (Fig. 6C,D). The body weight of these mice was maintained at the same level (Fig. EV5B). The data indicated that the tumor incidence of Sirt6 Floxed mice and Sirt6-LKO mice reached 91% (11/12) and 100% (12/12), respectively (Fig. EV5C). Of note, the tumor burden was significantly higher in Sirt6-LKO mice, especially in those tumor volumes less than 5 mm (Fig. EV5C). In addition, we also established the obesity-driven HCC mouse model, and the Sirt6-LKO in obesity mice markedly accelerated HCC development (Fig. 6E,F). The Sirt6 LKO mice gained more weight when compared with Sirt6 Floxed in this obesity model (Fig. EV5E). Of note, the spleen was enlarged and become calcified in Sirt6-LKO and ob/ob mice (Fig. EV5H), indicating the incapacity of an immune response to kill the tumor cells. Indeed, although the spleen weight was much heavier than in the Sirt6 Floxed ob/ob mice, there were fewer amounts of CD4+ and CD8+ T cells detected in Sirt6 LKO ob/ob mice (Fig. EV5I). Regarding ALT and

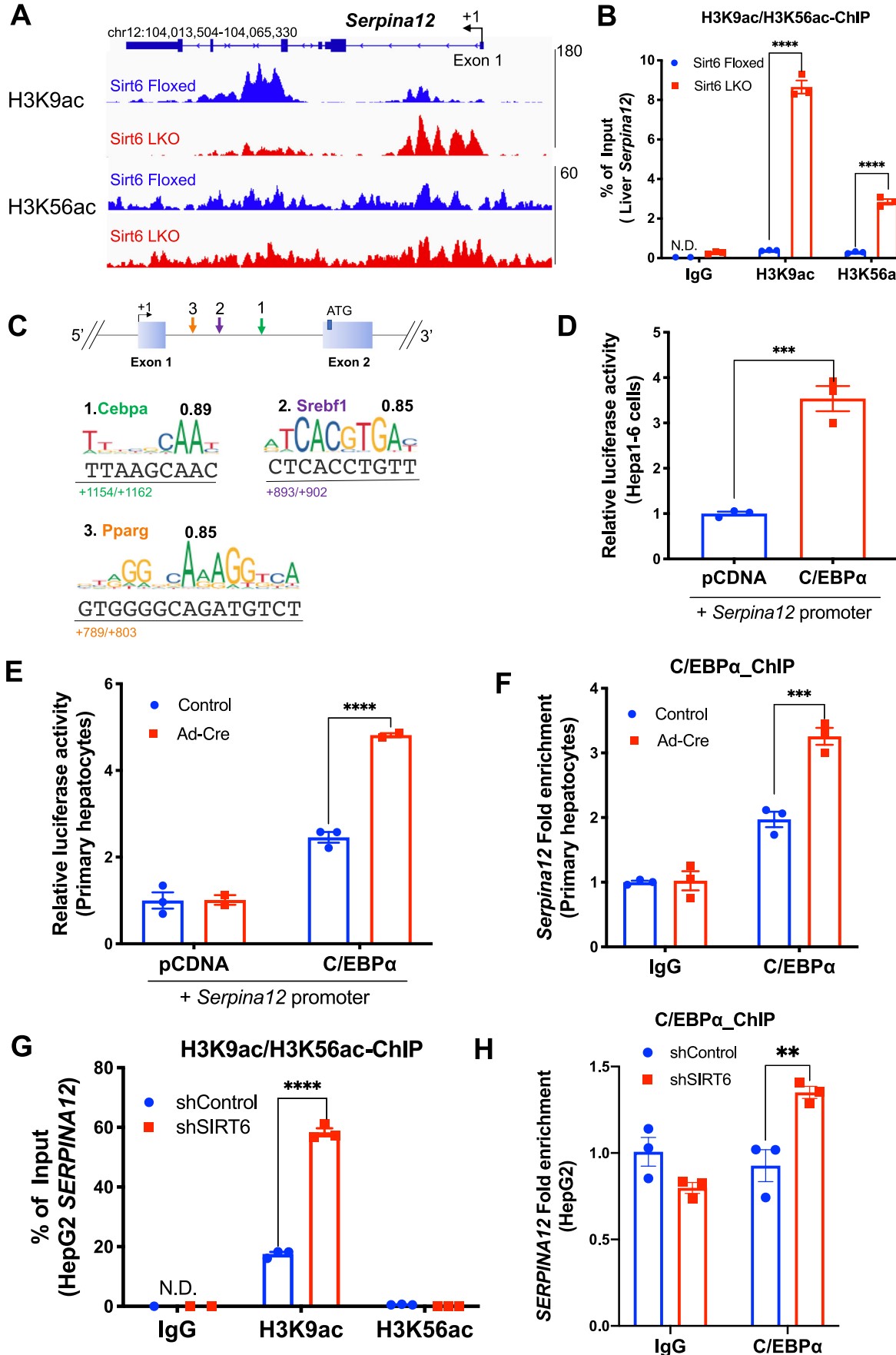

◄ **Figure 3.   The transcription factor CEBPα regulate Serpina12 expression.**

(A) IGV browser of images of read coverage across Serpina12 binding peaks in H3K9Ac and H3K56ac ChIP-seq from Sirt6 Floxed versus Sirt6 LKO mice liver. (B) H3K9ac and H3K56ac ChIP-qPCR experiments in Serpina12 genes. Samples were normalized to input and further normalized to IgG ChIP controls. $n = 3$ TR. (C) JASPAR analysis of the Serpina12 promoter bind transcription factors. (D) Luciferase activity after transfected promoter *m-serpina12-0.75k* and CEBPα in Hepa1-6 cells. $n = 3$ BR. (E) Luciferase activity after transfected promoter *m-serpina12-0.75k* and CEBPα in SIRT6 control or SIRT6 knock out primary hepatocytes. $n = 3$ BR. (F) CEBPα ChIP-qPCR experiments of Serpina12 gene expression either in SIRT6 control or SIRT6 knock out primary hepatocytes. $n = 3$ TR. (G) H3K9ac and H3K56ac ChIP-qPCR experiments in SERPINA12 gene expression either in SIRT6 control or SIRT6 knock down HepG2 cells. $n = 3$ TR. (H) CEBPα ChIP-qPCR experiments of SERPINA12 gene expression either in SIRT6 control or SIRT6 knock down HepG2 cells. $n = 3$ TR. Data information: For (B,D–H), error bars = SEM. Multiple *t*-test or two-way ANOVA. *$p < 0.05$, **$p < 0.01$, ***$p < 0.005$, ****$p < 0.001$. Source data are available online for this figure.

AST levels, no significant difference was observed in the liver between Sirt6 Floxed and LKO mice in the obesity model (Fig. EV5G) but appeared in the DEN-induced HCC model (Fig. EV5D).These data suggest that Sirt6 ablation promoted hepatocellular carcinoma. Interestingly, either the obesity-driven HCC or DEN animal models shared the pronounced lipid-rich environment in the tumor regions (Fig. 6A,C). Of note, Serpina12 expression was enhanced in the lipid-rich tumor region (Figs. 6G and EV5J), suggesting Sirt6 deficiency caused a lipid-rich environment through upregulated Serpina12 expression, thus accelerating liver cancer formation.

## Discussion

In this study, we have shown that upregulation of Serpina12 caused by Sirt6 deficiency plays a critical role in insulin signaling and lipogenesis to induce a lipid-rich environment in the liver. This lipid-rich environment thus serves as a soil that favors the tumorigenesis in Sirt6-LKO mice around two years of age in the liver. In the context of Sirt6 ablation in the liver, glycolysis and TG synthesis were enhanced which may facilitate tumor formation (Kim et al, 2010). Glycolysis has a central role in tumor cell proliferation and this process has been used in the past and present conception of the "Warburg Effect" (Vander Heiden et al, 2009). One of the proposed biosynthesis mechanisms for the Warburg Effect is de novo lipid synthesis which could support tumor cell proliferation (Liberti and Locasale, 2016). Sirt6 deficiency in the liver increases TG synthesis, providing a lipid-rich environment that promotes tumor formation. Still, the factors that mediate this lipogenic function in hepatocytes remain elusive. We propose that Serpina12, which is upregulated after Sirt6 ablation in the liver, plays a role in enhancing insulin signaling and promoting lipogenesis.

Serpina12 was first found as an adipokine in adipose tissue (Hida et al, 2005), and later, it was also found to be expressed in other tissues (Bluher, 2012). The effect of Serpina12 associated with insulin sensitization was identified in rats treated with insulin (Hida et al, 2005). Serpina12 deficient mice exhibit no obvious phenotype under normal conditions at least before 6-month of age (Nakatsuka et al, 2012). Serpina12 was demonstrated as a ligand binding with GRP78, a cell surface protein under the endoplasmic reticulum (ER) stress (Nakatsuka et al, 2012). Further study has demonstrated that it promotes the tumorigenic capacity of HCC stem cells through binding to GRP78, leading to the hyperactivation of Akt/β-catenin signaling (Yu et al, 2023). We found that Serpina12 is highly expressed in the liver after Sirt6 deficiency and

serves as a mediator of Sirt6 function in maintaining homeostasis in the liver. Although the function of Sirt6 or Serpina12 in insulin signaling has previously been shown separately (Xiao et al, 2010), our work has uncovered an unknown mechanism for Sirt6 in regulating the process of insulin signaling through Serpina12 in the liver. Thus, our studies advance upon previous work and demonstrate that Sirt6 deficiency enhances insulin signaling through the upregulation of Serpina12 in primary hepatocytes.

In the liver, Sirt6 deficiency caused by liver specific knockout promotes lipogenesis (Kim et al, 2010). However, it is unclear whether or not the direct effect of increased expression of Serpina12 after Sirt6 deficiency would cause lipogenesis in hepatocytes. Our results have demonstrated that both under high glucose and oleic acid stimulated lipogenesis conditions, this effect could be reduced in the Serpina12 knockdown group, suggesting Serpina12 is involved in lipogenesis after Sirt6 deficiency. As increased Serpina12 also enhances insulin signaling, which is reflected by the IRS1 phosphorylation level after Sirt6 deficiency, Serpina12 might also serve as the consequence of lipogenesis in the liver. As proof for this, we showed that knockout Serpina12 in the liver through hydrodynamic injection in the tail veil abolished the fatty liver phenotype after Sirt6 deficiency, thus providing a possibility of Serpina12 in regulating lipid homeostasis in mice. We also showed that Serpina12 promotes lipogenesis by affecting lipid-related gene expression, such as *Fasn*, *Elovl6*, and *Scd1*. The regulation of Serpina12 on these downstream genes may be through enhanced insulin signaling, leading to activating of these genes. Further study may be needed to investigate this issue. Insulin-dependent signaling pathways are often overactivated in human HCC due to the overexpression of signaling components and loss of negative regulators (Alqahtani et al, 2019; Enguita-German and Fortes, 2014). As a result, hyperinsulinemia can directly promote cancer cell metabolism, proliferation, and survival, potentially contributing to HCC development and progression (Chettouh et al, 2015). We have tested the effect of insulin signaling in HCC and the data indicated that insulin signaling plays a crucial role in the development of hepatocellular carcinoma (HCC) by promoting cell proliferation (Appendix Fig. S3).

Sirt6 regulates Serpina12 through epigenetic modification. The ChIP-seq and ChIP-qPCR results revealed that significantly enriched binding appeared in H3K9ac and H3K56ac after Sirt6 deficiency. These two sites of modification in acetylation represent an open chromatin structure to allow more transcription factors to regulate gene expression (Michishita et al, 2008, 2009). In this perspective, we found that a CCAAT binding site, which is recognized by Cebp/α (Ramji and Foka, 2002), exists in the Serpina12 promoter. Our results demonstrate that Cebp/α could

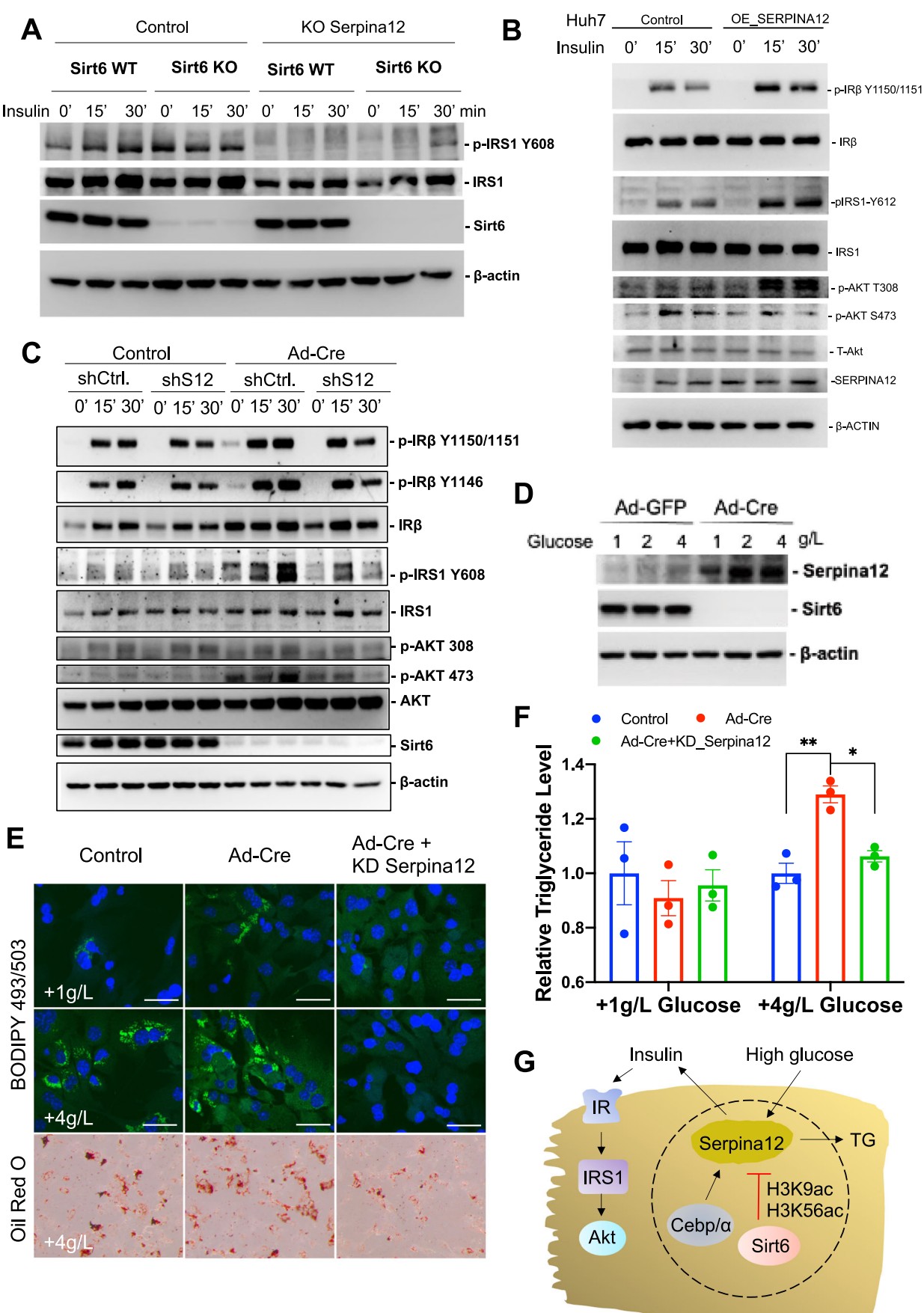

**Figure 4. Serpina12 promote lipogenesis and enhance insulin signaling.**

(A) Western blot analysis of insulin signaling following insulin treatment at 0, 15 and 30 min in control or sgSerpina12 from either control or adeno-Cre in primary hepatocytes. (B) Western blot analysis of insulin signaling following insulin treatment at 0, 15 and 30 min in control or overexpression of Serpina12 in Huh7 cells. (C) Western blot analysis of insulin signaling following insulin treatment at 0, 15 and 30 min in control or Ad-Cre with or without knockdown Serpina12 in primary hepatocytes. (D) Western blot analysis of Serpina12 protein levels in different glucose concentration. (E) Lipogenesis induced by glucose in primary hepatocytes. Cells were treated with 1 g/L or 4 g/L glucose 12 h followed by staining with fluorescent dye BODIPY 493/503; Oil red O staining of control, knockout SIRT6 with or without knockdown Serpina12 primary hepatocytes. Scale bars: 50 μm. (F) Triglyceride (TG) level in primary hepatocytes treated with 1 g/L or 4 g/L glucose in control, knockout SIRT6 with or without knockdown Serpina12 groups. $n = 3$ BR; error bars = SEM. Two-way ANOVA. $*p < 0.05$, $**p < 0.01$. (G) The cartoon summarized the sirt6 regulated serpina12 expression. Source data are available online for this figure.

increase Serpina12 expression, and this increase is abolished after Cebp/α knockdown in HepG2 cells, indicating both Sirt6 deficiency and Cebp/α are required for its expression.

Sirt6-LKO mice develop fatty liver, which is a well-known driver for HCC (Younossi et al, 2015). However, whether it will progress to liver cancer remains unknown. Sirt6 has been shown to act as a cancer suppressor in many studies (Bhardwaj and Das, 2016; Kugel et al, 2016; Min et al, 2012). The transcription factor AP-1 component c-Jun promoted cell survival during cancer initiation by inhibiting c-Fos-induced Sirt6 expression (Min et al, 2012; Xiao et al, 2012). Specifically, Sirt6 repressed oncogenic Survivin expression through histone H3K9 deacetylation and reduced the p65/NF-κB binding to its promoter, thus augmenting apoptosis (Min et al, 2012). Another study has found that UBE3A targets SIRT6 at K160, resulting in its ubiquitination and degradation. SIRT6 downregulation elevates the ANXA2 level and promotes UBE3A-mediated tumorigenesis in HCC (Kohli et al, 2018). Our results have also shown that Sirt6 is a tumor suppressor in the liver, as Sirt6-LKO mice spontaneously developed tumors at two years old. Besides, we found that Sirt6-LKO mice manifest features of steatosis in the tumor region, suggesting this alteration could drive the development of carcinomagenesis (Salomao et al, 2010). Another well-known deleted driver in human cancer is *PTEN* (Lee et al, 2018; Li et al, 1997; Song et al, 2012). About a half percentage of HCC and two third of cholangiocarcinoma were found to lose PTEN expression (Sze et al, 2011; Yothaisong et al, 2013). In the Pten-deleted Sox9+ cells, liver cancer was found with mixed-lineage tumors of both HCC and intrahepatic cholangiocarcinoma (Chen et al, 2021). Similarly, our findings showed that Sirt6-LKO also results in a mixed tumor pattern, which is accelerated in a DEN-induced liver cancer model (Dapito et al, 2012; Yoshimoto et al, 2013). To offer further insights into lipid accumulation leading to tumorigenesis, we used a leptin-deficient (obese) mouse model and found that loss of Sirt6 accelerates tumor formation in the liver, accompanied by reduced CD4+ and CD8+ T cells in the liver. The reduced immunity could lead to the failure to eliminate cancer cells in the liver, thus promoting the tumor formation. These data further support our observation that Sirt6 LKO develops fatty liver and carcinogenesis at the old stage due to lipid accumulation in the liver. In summary, this study provides a potential mechanism for the ablation of Sirt6, which triggers an increase in Serpina12 upregulation to induce a "lipid-rich" environment that favors liver cancer development (Fig. 7).

We observed that the immune response related pathway was decreased in Sirt6 LKO mice using RNA-seq analysis. The most significant downregulated pathway is Cellular Response to Interferon-Beta. Interferon beta binds to its receptor on the cell surface, leading to the activation of JAK-STAT signaling pathway to against viral infection. These pathways were downregulated, suggesting the weaker activity of the immune cells in the Sirt6 LKO mice liver. Although we found that the number of Kupffer cells has no obvious change in the liver, T cells could also affect immune processes. In Sirt6 LKO mice with obesity background, we have found reduced CD4+ and CD8+ T cells in the liver. The reduced immunity could lead to the failure to eliminate cancer cells in the liver, thus promoting the tumor formation. Meanwhile, the reduced immunity could also lead to chronic inflammation and the accumulation of damaged cells in the liver. Therefore, it is possible that boosting the immune system may help in preventing or treating liver diseases in this background. However, the relationship between the immune system and cancer is complex, and enhancing the immune system alone may not be effective to treat liver cancer in clinical condition. Further study is needed to illustrate the effect in this case.

## Methods

See Table 1.

### Animal experiments

Sirt6 liver conditional knockout mice were generated according to the procedure described earlier (Kim et al, 2010). For single sgSerpina12 delivery, the constructed vector pX330 or pX330-sgSerpina12 (25ug) was dissolved in 1.8 ml sterile PBS and injected into Sirt6 LKO mice at one-month-old age within 5 s via tail vein. For DEN treatment, mice were challenged by the DEN intraperitoneally (25 mg/kg) on a postnatal day 14. Leptin mutation mice (The Jackson Laboratory, stock number: 000632) were crossed with Sirt6 liver conditional knockout mice to generate the obesity-driven liver cancer model. All animal experiments were guided by the rules of the University of Macau Animal Care and Use Committee (UMARE-015-2019).

### Isolate primary hepatocytes

Sirt6$^{co/co}$ mice of the age around 4 weeks were used to isolate mouse primary hepatocytes. A modified two-step collagenase perfusion procedure was used to isolate primary mouse hepatocytes (Charni-Natan and Goldstein, 2020). Briefly, the liver tissue was perfused with perfusion buffer I to remove residual blood cells, followed by perfusion buffer II (perfusion buffer I plus 1 mM EDTA, 5 mM

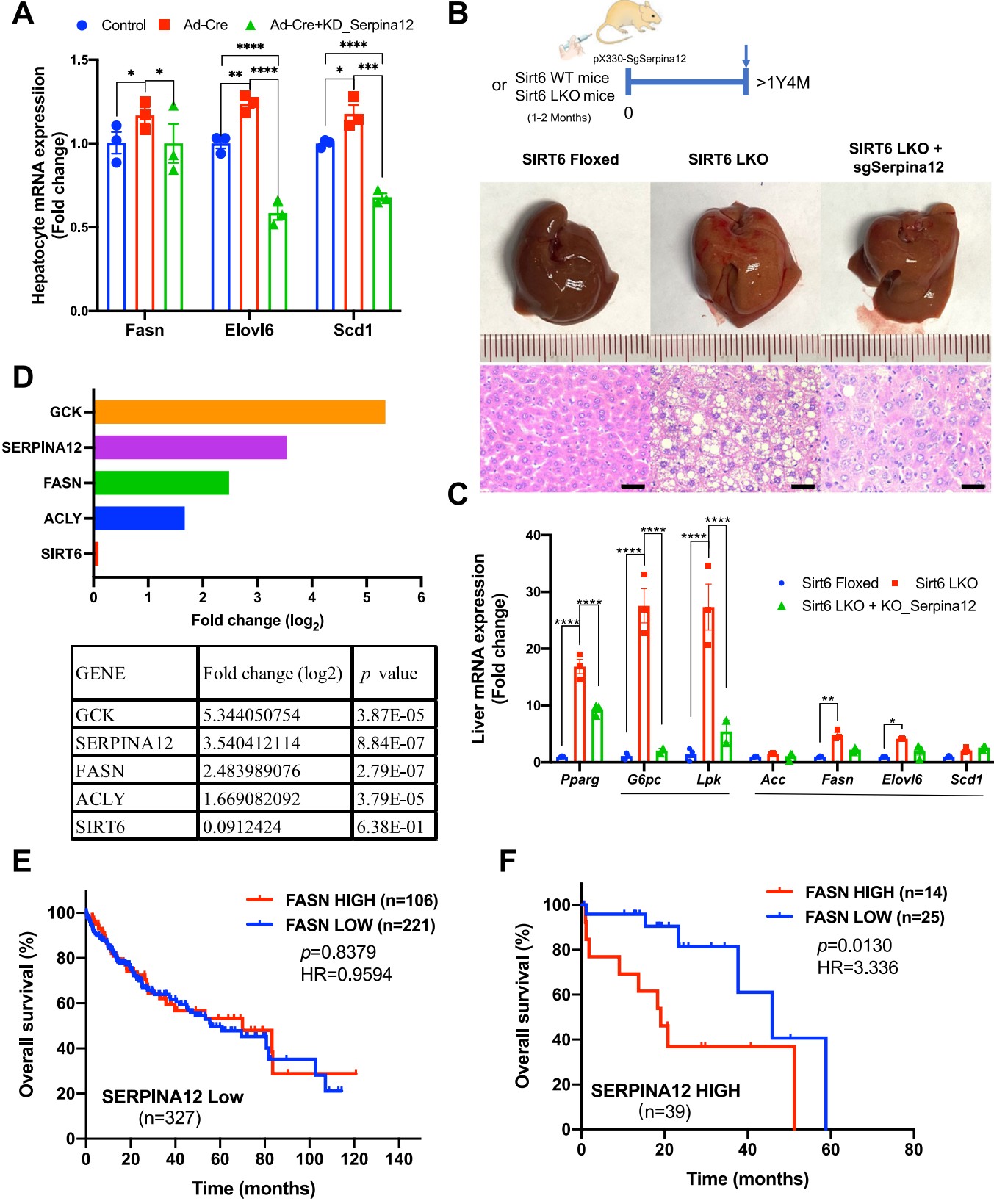

**Figure 5. Knockdown Serpina12 alleviate Sirt6 deficient caused fatty liver in mice.**

(A) qPCR showed the relative lipid-related genes expression in primary hepatocytes after knockout Sirt6 or double knockout Sirt6 and Serpina12. $n = 3$ TR. (B) Morphology of Sirt6 Floxed and Sirt6 LKO with or without knockout Serpina12 mice liver at 1-year and 4-month-old age; H&E staining of Sirt6 Floxed and Sirt6 LKO with or without knockout Serpina12 mice liver sections. Scale bars:10 μm. (C) qPCR showed the relative glucose and lipid-related genes expression in liver. $n = 3$ BR. (D) Comparison of relative expression of GCK, SERPINA12, FASN, ACLY, and SIRT6 in healthy human liver compared to NASH (GSE89063) showing fold change (log2), and statistical significance ($P$ value) of fold change for each gene. (E,F) Overall HCC survival data obtained from the TCGA database for tumors showing SERPINA12 low expression (above the mean, (E)) versus SERPINA12 high expression (below the mean, (F)) with either FASN high or FASN low expression. Data information: For (A) and (C), error bars = SEM. Two-way ANOVA. $*p < 0.05$, $**p < 0.01$, $***p < 0.005$, $****p < 0.001$. Source data are available online for this figure.

CaCl2) perfusion. The tissue was then perfused with perfusion buffer III (perfusion buffer I plus 1 mg/mL collagenase type IV, 5 mM CaCl2). All buffer solutions were prewarmed to 37 °C before the isolation process. The hepatocyte suspension was collected and washed with HBSS, then filtered through a 40 μm Nylon cell strainer. After centrifugation several times and purifying with 90% percoll solution, the isolated hepatocytes were then seeded on cell culture plates in the Williams' medium E.

## RNA abstract, qPCR, RNA-seq, and data analysis

The liver of Sirt6 Floxed and Sirt6 LKO mice ($n = 3$) were used to abstract total RNA. Briefly, Trizol, chloroform, 2-propanol and 75% ethyl alcohol were used in this method. RNA samples were sent out to sequence by the company. After normalizing the data, genes were annotated according to NCBI. KEGG pathway analysis for the differentially expressed genes was used in the online software (DAVID 2021). After quantified by Nanodrop, a total 1ug RNA were used to reverse transcription to obtain the cDNA (Qiagen). cDNA was diluted 5 times to perform the qPCR procedure. mSerpina12 Fwd-TTGCTC GACACAACATGGAAT, Rev-CGTCCCAGTTTGACATCTCTTT; mCyp2b10 Fwd-AAAGTCCCGTGGCAACTTCC, Rev-TTGGCTCA ACGACAGCAACT; mGal3st1 Fwd-CTGCTGCCAAAGAAGCC CT, Rev-AGGCCATGTTGGGATACAGTG; mSirt6 Fwd-ATGTCG GTGAATTATGCAGCA, Rev-GCTGGAGGACTGCCACATTA; mOrm3 Fwd-GAACTACACACGGTTCTCATCAT, Rev-CATTTG CCAGATAGCCAGCTC; 18S Fwd-AGTCCCTGCCCTTTGTACAC A, Rev- CGATCCGAGGGCCTCACTA.

## Chromatin Immunoprecipitation (ChIP) and ChIP-seq

Liver tissues were cut into small pieces and snapped frozen in liquid nitrogen. Weigh around 25 mg of liver samples and powder them on liquid nitrogen. Then the powder was collected in the 50 ml falcon tube with 1% formalin in PBS for 10 min at room temperature (RT) by gently shaking. The glycine was added to the final concentration at 0.125 M to neutralize the formalin for 5 min at RT. After washing twice with PBS, the samples were lysed with lysis buffer on ice. Lysates were then sonicated at 4 °C in the cold room with the pulse 30 s on and 30 s off for 20 min. DNA agarose gel was used to confirm the size of fragments (200–500 bp). The supernatant was collected after centrifugation at 14,000 rpm at 4 °C for 10 min. Then the samples were divided into five parts. One part is for Input, and the other four parts are for antibodies against IgG, H3K9ac, and H3K56ac (Normal Rabbit IgG -CST 2729 s; Histone H3 (acetyl K9)-Abcam ab4441; H3K56ac-Invitrogen PA5-40101). After incubating samples overnight at 4 °C in rotation, samples were washed by wash buffer and collected by elution buffer. Then the samples were de-crosslink by adding RNAse A and

protease K and incubated at 65°C overnight. DNA was purified using the PCR purification kit according to the manufacturer's instructions. The DNA concentration was quantified by Picogreen. The DNA was sent to company for library construction and sequencing. The sequencing reads were aligned to the mm10 reference genome and used to identify H3K9ac or H3K56ac associated differential peaks between Sirt6 Floxed and Sirt6 LKO livers.

## Lentiviral production and transduction

HEK293/17 cells were used to package lentiviral particles. For the purpose of knockdown SIRT6 in primary hepatocytes and HepG2 cells, pLKO.1-shSIRT6 lentiviruses were directly added into the cells with the existence of 10 μg/mL polybrene. After transduction 72 h, cells were used for treatment as well as collected to test the knockdown efficiency by western blot.

## Western blot analysis

Cell pellets were resuspended and lysate by RIPA buffer containing PMSF, protease and phosphatase inhibitors. After centrifugation at 14,000 rpm for 10 min at 4 °C, the supernatant was collected to measure the protein concentration. The antibodies against p-AKT, AKT, p-IR, Serpina12, SIRT6, and Actin were used to immunoblot after running the SDS-PAGE gels.

The following antibodies were used: (SirT6-CST 12486 S; Phospho-Akt (Ser473)-CST 4060 s; Akt (pan)-CST 4685 s; Phospho-IGF-I Receptor β (Tyr1135/1136)/Insulin Receptor β (Tyr1150/1151)-CST 3024 s; Insulin Receptor β-CST 3025 s; Anti-phospho-IRS1 (Tyr608) mouse/ (Tyr612) human Antibody-Sigma 09-432; IRS1-CST 2382 s; Recombinant Anti-Vaspin antibody-Abcam ab267470; Anti-Vaspin (human)-Adipogen AG-20A-0045-C100; C/EBP alpha polyclonal antibody-Thermo PA5-77911; β-actin-CST 3700 s).

## Lipid staining

Cells were washed with PBS three time and incubated with 2 μM Bodipy (Invitrogen, D3922) for 15 min at 37 °C and protect from light. After wash twice with PBS, fixed the cells with 4% PFA for 30 min at RT. Wash three time and stained with DAPI (0.1 μg/ml).

## Cloning

H_SERPINA12_0.8k promoter fragment and m_Serpina12_0.75k promoter fragment were amplified by Q5 master mix PCR procedures. And then the fragments were purified by PCR and double digestion with restriction enzymes MluI and BglII. And then the promoter fragments were constructed into the pGL3-basic

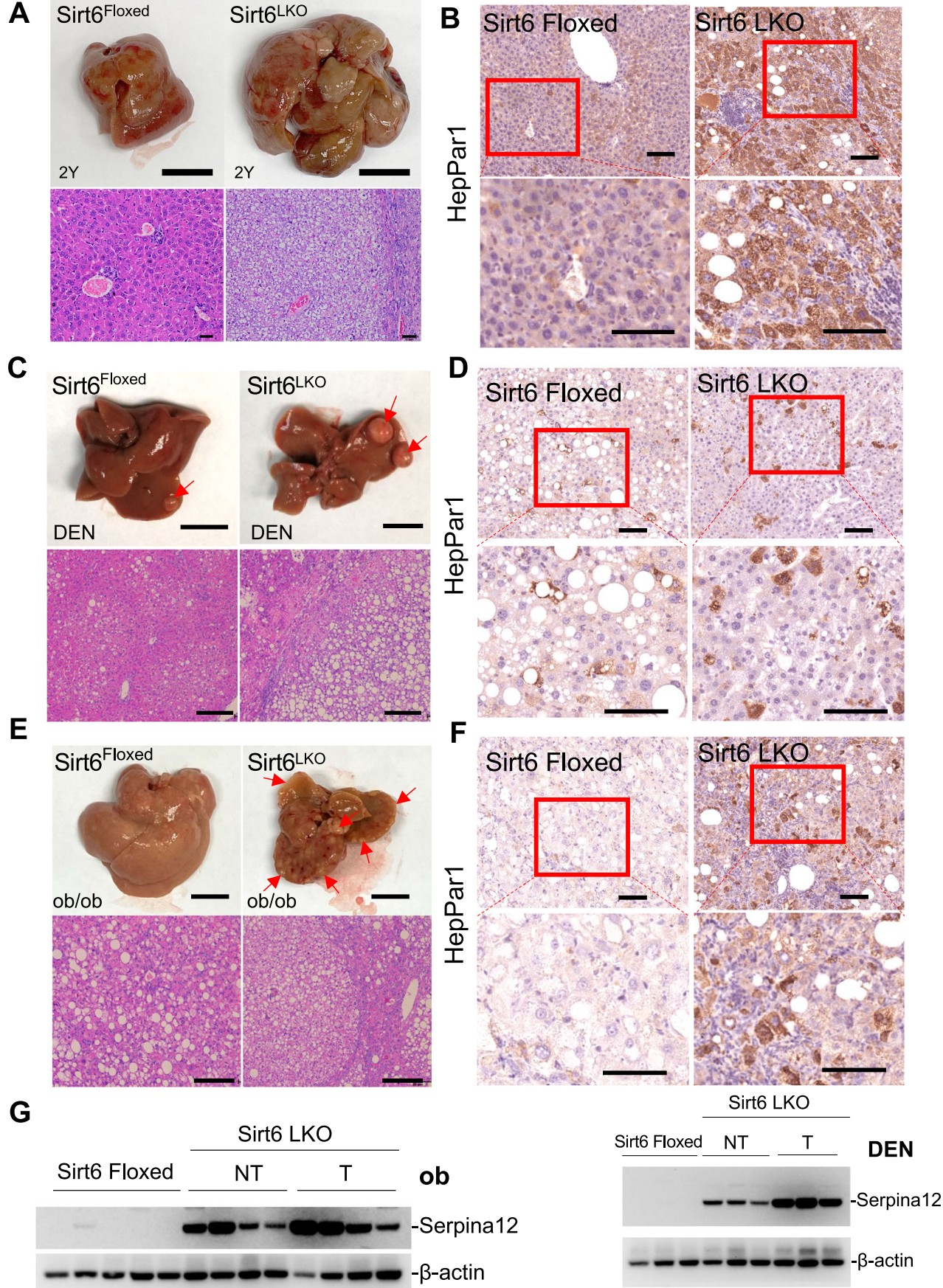

**Figure 6.   Sirt6 deficient caused lipid-rich environment to accelerate tumor formation in mice.**

(A) Liver morphology of Sirt6 Floxed or Sirt6 LKO mice liver of 2-year-old age, Scale bars:10 mm; H&E staining of Sirt6 Floxed and Sirt6 LKO mice, Scale bars:10 µm. (B) IHC staining of HCC marker, HepPar1, in Sirt6 Floxed and Sirt6 LKO mice at 2-year-old age, Scale bars: 100 µm. (C) Liver morphology of Sirt6 Floxed and Sirt6 LKO with DEN injection mice at 7-month-old, Scale bars:10 mm; H&E staining of the indicated liver sections, Scale bars: 50 µm. (D) IHC staining of HCC marker, HepPar1, in Sirt6 Floxed and Sirt6 LKO mice with DEN injection mice, Scale bars: 100 µm. (E) Liver morphology of Sirt6 Floxed and Sirt6 LKO with ob/ob mice at 7-month-old, Scale bars:10 mm; H&E staining of the indicated liver sections, Scale bars: 50 µm. (F) IHC staining of HCC marker, HepPar1, in Sirt6 Floxed and Sirt6 LKO with ob/ob mice, Scale bars: 100 µm. (G) Western blot analysis of Seprina12 in Sirt6 Floxed and Sirt6 LKO with ob/ob or DEN injection mice liver tissue. Source data are available online for this figure.

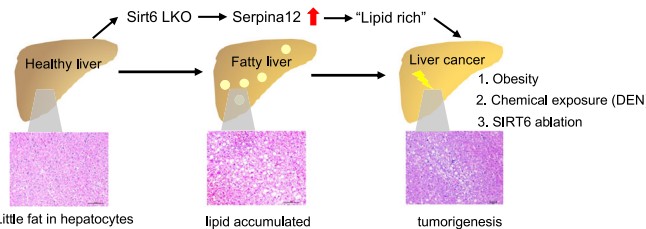

**Figure 7.   Schematic model illustrating lipid accumulated in Sirt6 LKO developed to liver cancer.**

Sirt6 deficiency in the liver causes a lipid-rich environment via upregulation of Serpina12 and promotes tumorigenesis.

vector with further confirmed with Sanger sequence. The lentiviral shRNA plasmid pLKO.1 targeting human SIRT6, mouse Sirt6, human SERPINA12, mouse Serpina12 and human CEBP/α were constructed followed by the instruction of the Addgene online websites. Mouse sgSerpina12 were constructed into either pX330 or lentiCRISPR v2 plasmid following the Addgene instruction.

## Luciferase assay

The cells were transfected with indicated plasmids by using Lipofectamine 3000 or Lipofectamine LTX (Thermo Fisher Scientific). pRL-TK was co-transfected to the cells to normalize transfection efficiency. The cells were transfected with Serpina12 promoter luciferase reporter after with or without knockdown of SIRT6 for 48 h. The cells were harvested 24 h post-transfection to measure luciferase activity. the measurement was using a dual luciferase reporter assay system following the indication of the protocol (Promega).

## TG measurements

The hepatocytes were harvested and used the triglyceride quantification assay kit (Abcam, ab65336) to extract the TG and followed measured by the fluorescence (Ex/Em = 535/587 nm).

## H&E staining

The tissues were cut and fixed in 10% formalin for 24 h and transferred into 70% ethanol for the further procedure after washing twice with PBS. The embedded tissues were dissected to 5µm using the microtome. The sections were first deparaffinized and rehydrated with gradient ethanol and then stained with hematoxylin for 30 s. After washing with tap water, the sections

were stained with 0.5% eosin for 2 min and further with dehydration and mounted with DPX.

## Immunoprecipitation

Cells were pellet down at 1000 rpm for 5 min at 4 °C. In brief, cells were lysed on ice using IP Buffer (Thermo Scientific™,Pierce™ IP Lysis Buffer, 87787) with protease inhibitor for about 1 h. The lysate was sonicated at 20% output for 10 pulses, and then spin the lysates at 4 °C at 13,000 RPM for 30 min. Supernatant (1 mg total protein) was transferred to a new tube and precleared with protein A/G PLUS-Agrose beads (sc-2003) at 4 °C for 1 h on a rotator. Beads were removed by centrifugation and transfer of cleared sample to new tubes. Samples were incubated overnight at 4 °C with 2 µg of antibody (SirT6-CST 12486 s; C/EBP alpha-Invitrogen PA5-77911). Next day, spin the mixture at 6000 rpm for 6 min at 4 °C. Discard the supernatant and wash the beads with 300 mM NaCl in lysis buffer.Final resuspension with 100 µl IP buffer.

## Bioinformatic analysis

Expression data from dataset GSE89063 were downloaded from the Gene Expression Omnibus (GEO) database, and BioJupies web application (Torre et al, 2018) was used to perform bioinformatic analysis of SERPINA12 expression and lipid-associated genes between LGLI (GSM2358163; GSM2358164; GSM2358165; GSM2358169; GSM2358170; GSM2358171; GSM2358175; GSM2358176 and GSM2358177) versus NASH (GSM2358166; GSM2358167; GSM2358168; GSM2358172; GSM2358173; GSM2358174; GSM2358178 and GSM2358179). Linear regression correlation analysis between SERPINA12 and GCK, FASN and ACLY was performed using BioVinci (Bioturing, San Diego, CA, USA).

Liver hepatocellular carcinoma (LIHC) patient's survival analysis was performed using cBioPortal (Cerami et al, 2012; Gao et al, 2013) and data from the TCGA PanCancer Atlas ($n = 372$ patients). LIHC patients were classified into different groups according to the SERPINA12 and FASN mean expression value. By the Kaplan–Meier method, we obtained the overall survival and comparison between curves was made using the log-rank test. Data obtained from cBioPortal database do not require ethical approval.

## Statistical analysis

Data were analyzed using GraphPad Prism software. For comparison of different samples, multiple $t$-test or two-way ANOVA test were used to analyze the statistical difference. Statistically

**Table 1.  Reagents and tools table.**

| Reagent/Resource | Reference or Source | Identifier or Catalog Number |
|---|---|---|
| **Antibodies** | | |
| Rabbit anti-SirT6 | CST | Cat#12486 S |
| Rabbit anti-Phospho-Akt (Ser473) | CST | Cat#4060 s |
| Rabbit anti-Akt (pan) | CST | Cat#4685 s |
| Rabbit anti-Phospho-IGF-I Receptor β (Tyr1135/1136)/Insulin Receptor β (Tyr1150/1151) | CST | Cat#3024 s |
| Rabbit anti-Insulin Receptor β | CST | Cat#3025 s |
| Rabbit anti-phospho-IRS1 (Tyr608) mouse/ (Tyr612) human Antibody | Sigma | Cat#09-432 |
| Rabbit anti-IRS1 | CST | Cat#2382 s |
| Rabbit anti-Recombinant Anti-Vaspin antibody | Abcam | Cat#ab267470 |
| Mouse Anti-Vaspin (human) | Adipogen | Cat#AG-20A-0045-C100 |
| Rabbit anti-C/EBP alpha polyclonal antibody | Thermo | Cat#PA5-77911 |
| Rabbit anti-Histone H3 (acetyl K9) | Abcam | Cat#ab4441 |
| Rabbit anti-H3K56ac | Invitrogen | Cat#PA5-40101 |
| Rabbit anti-Normal Rabbit IgG | CST | Cat#2729 s |
| Mouse anti-β-actin | CST | Cat#3700 s |
| **RT-primers** | **Forward** | **Reverse** |
| Cyp4a10 | TTCCCTGATGGACGCTCTTTA | GCAAACCTGGAAGGGTCAAAC |
| Serpina12 | TTGCTCGACACAACATGGAAT | CGTCCCAGTTTGACATCTCTTT |
| Ptgds | TGCAGCCCAACTTTCAACAAG | TGGTCTCACACTGGTTTTTCCT |
| Ldhb | CATTGCGTCCGTTGCAGATG | GGAGGAACAAGCTCCCGTG |
| Cyp2u1 | TCCAAGGGGTTCACCATTCCC | CGATGAGGACAGAAGTCGTCT |
| Cyp2b10 | AAAGTCCCGTGGCAACTTCC | TTGGCTCAACGACAGCAACT |
| Csrp3 | GGGGGAGGTGCAAAATGTG | CAGGCCATGCAGTGGAAACA |
| Acnat2 | AAGCGGGAACAGATTCAAGAAG | ACGAAATTCAACTAGACCCCCA |
| Atp8b4 | GAGAAGTTCCAGTATGCGGAC | TGACAGCCGTCATCGAGATCA |
| Ces1b | ACTGCCACAGGGTGTTCAAAT | GCTCTTTGCCCAATATGGGGA |
| Cxcl17 | AGGTGGCTCTTGGAAGGTG | GGTGACATCGTTTGAGAAATTGC |
| Gal3st1 | CTGCTGCCAAAGAAGCCCT | AGGCCATGTTGGGGATACAGTG |
| Gstt3 | GGATGGGGACTTCGTCTTGG | TCAGGAGGTACGGGCTGTC |
| Marveld3 | GGAGAGGTGTGGTGCAGATAA | GGTGCCCTCAAAAGGTGAGTAA |
| Mmd2 | AGTATGAACACGCAGCAAACT | TCCCAGTCGTCATCGGACA |
| Lmod3 | ATGGTGAACAAGCCAACAGAG | GCCTCGCACTTACCTTCAGA |
| Apba2 | TCACCTACTACATCCGCTACTG | CCTCCTGACACTCATCGGT |
| Tmigd1 | GGTGTCCAAGCATCTCTGGAA | CCATTGTCACTTTCGTTGATGGG |
| Grid1 | GGATATGCCAGTGCGTTACG | TTGAGGCTCAGGTCTGATACA |
| Slc13a4 | GGAAGCTGCTATTGGTCATCTG | GGTCACAAGCAACACGTAAGC |
| Olfr541 | ATGCTAAGACTGAACCAGACAGA | CAGGATGCTGCCAATGATGT |
| Pax8 | ATGCCTCACAACTCGATCAGA | ACAATGCGTTGACGTACAACTT |
| Vtcn1 | CTTTGGCATTTCAGGCAAGCA | TGATGTCAGGTTCAAAAGTGCAG |
| Klf17 | AATAAGGAACAGGCTATGCACC | GTGGCTGATGAAATCCGCTG |
| Tmprss4 | CAACCCCTCAACAACCGTGAT | CTCAGCAGCACTGCAATGAT |
| Stap1 | TCACTGCTCTGCCCCTGTA | AGTAAGGCACACGAGGTCTATTA |
| Gm6614 | CCTGCGACCCAATTTACGTG | GCAGAATGCTAGGTATGCAGC |
| Ccdc162 | CAGATGTGTCTCTGTGGGCTA | TTGCCATTCGGTAGAAGCCTT |

**Table 1.** (continued)

| Reagent/Resource | Reference or Source | Identifier or Catalog Number |
|---|---|---|
| Sema5b | AGCACGTCACAACACACTCTG | GGTACAAGTTCTCTTGCGGAC |
| Corin | GCTGGTGACTGCTAACTTGCT | CCCATCAGTGACCAAAGGTTC |
| Lama3 | CTGTGACTACTGCAATTCTGAGG | CAAGGTGAGGTTGACTTGATTGT |
| Orm3 | GAACTACACACGGTTCTCATCAT | CATTTGCCAGATAGCCAGCTC |
| Sema3b | GTAGCAGGGCTAGGGGATACT | AAGGCTTCATAACAGCAGGTC |
| Serpina5 | AGAAGAAGGCTAAAGAGTCCTCG | CTCATAGACACGCTCAAGGGG |
| Dsg1c | CATCCACTGAGAAACCTGTGAC | CTGTGGCCCTTCTACAGCC |
| D630033O11Rik | ATGTCCCGTTAGCCCTTCCA | GCAGAAGAGCATCACCCAGTT |
| Slco1a4 | GCTTTTCCAAGATCAAGGCATTT | CGTGGGGATACCGAATTGTCT |
| Hectd2 | GCAGCTTTGTCTCGACTGTCA | ACTTCAGATGGTAACGCTTCCA |
| Pcp4l1 | GCATTTGGGGCAGCATTAAGC | CTTCCCTTTTTCCTCAGGGTC |
| Tfcp2l1 | CAGCCCGAACACTACAACCAG | CAGCCGGATTTCATACGACTG |
| Adgrv1 | GGATGACGCAGGTCCTTTTA | CCTCCAGGGTCATCATTTTC |
| Nat8f7 | GCAACACCAGCTTCACAAGA | GCAATCTTCCTCTGCCTTTG |
| Mup8 | AGAGATGAAGAGTGCTCGGAAT | GTTTCCCCATCCTTTTCGTT |
| Mup12 | AGAGATGAAGAGTGCTCGGAAT | GTTTCCCCATCCTTTTCGTT |
| Selenbp2 | CGCATATTTGTGTGGGACTG | CAACATCCAGCCCTTCACTT |
| Fam186a | ACTGGGAAAGGGTACGAGGT | TCTTAAAGCTGGGGATCGAA |
| Trp63 | GTCAGCCACCTGGACGTATT | CTCATTGAACTCACGGCTCA |
| 18S | AGTCCCTGCCCTTTGTACACA | CGATCCGAGGGCCTCACTA |
| sgRNA sequence | Forward | Reverse |
| hSERPINA12_1 | CACCGCTGATCGAGAATATAGACCC | AAACGGGTCTATATTCTCGATCAGC |
| hSERPINA12_2 | CACCGGAATTATAAAGCTTTGAGCG | AAACCGCTCAAAGCTTTATAATTCC |
| hSERPINA12_3 | CACCGGTGCCCATGATGTTCCGTAG | AAACCTACGGAACATCATGGGCACC |
| hSERPINA12_4 | CACCGTCTTCCAAAAACTTACGCTG | AAACCAGCGTAAGTTTTTGGAAGAC |
| mSerpina12_1 | CACCGATGGTAAAAAGCATAGACCC | AAACGGGTCTATGCTTTTTACCATC |
| mSerpina12_2 | CACCGGATAGGAAGATGTTTCCCTG | AAACCAGGGAAACATCTTCCTATCC |
| mSerpina12_3 | CACCGGAGGACACAAAGATGAACTT | AAACAAGTTCATCTTTGTGTCCTCC |
| mSerpina12_4 | CACCGTATGTATGATTCTCCAGTCA | AAACTGACTGGAGAATCATACATAC |
| mSerpina12_5 | CACCGCCATGTCGTACAAGCCCCTT | AAACAAGGGGCTTGTACGACATGGC |
| mSerpina12_6 | CACCGTGTTGTGTCGAGCAAGCTGC | AAACGCAGCTTGCTCGACACAACAC |
| mSerpina12_7 | CACCGGTTTCCCTGAGGGCTATTGC | AAACGCAATAGCCCTCAGGGAAACC |

significant difference was labeled in the figures. A $p$ value of <0.05 was considered statistically significant difference. $*p < 0.05$, $**p < 0.01$, $***p < 0.005$, $****p < 0.001$.

## Data availability

All sequencing data have been deposited in the Sequence Read Archive (SRA) under the bioproject number RNAseq data, Sequence Read Archive PRJNA1037177 ChIPseq data, Sequence Read Archive PRJNA1037177.

## Peer review information

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

## Acknowledgements

We would like to thank Dr. H. Sun (Faculty of Health Sciences, University of Macau) and Dr. K. Miao (Faculty of Health Sciences, University of Macau) for the technical assistance during experiments. We thank Dr. John T. Heiker for providing the Serpina12 expressing plasmid. We also thank the members of the C.X.D. and X.X. laboratories for helpful advice and discussions. We are grateful for the support of the Animal Research Core and Biological Imaging and Stem Cell Core of the Faculty of Health Sciences, University of Macau. This work was supported by Chair Professor Grant (CPG2023-00031-FHS) from University of Macau, Macau SAR, China; Multiyear research grant (MYRG2022-00181-FHS, MYRG2020-00076-FHS) from University of Macau, Macau SAR, China; Macao Science and Technology Development Fund (FDCT: 0092/2020/AMJ, 0004-2021-AKP, 0007/2021/AKP, 0009/2022/AKP, 0065/2021/A, 0058/2022/A1, 0034/2019/AGJ and 0054/2023/RIA1); and Natural Science Foundation of China: 82030094, 81602587 and 81672603.

## Author contributions

**Licen Li**: Conceptualization; Formal analysis; Investigation; Visualization; Methodology; Writing—original draft; Writing—review and editing. **Jianming Zeng**: Investigation; Visualization; Methodology. **Xin Zhang**: Investigation. **Yangyang Feng**: Investigation. **Josh Haipeng Lei**: Investigation. **Xiaoling Xu**: Resources; Methodology. **Qiang Chen**: Supervision; Funding acquisition. **Chu-Xia Deng**: Conceptualization; Resources; Supervision; Funding acquisition; Methodology; Writing—original draft; Project administration; Writing—review and editing.

## Disclosure and competing interests statement

The authors declare no competing interests.

# Expanded View Figures

**Figure EV1. Sirt6 ablation induces formation of fatty liver accompanied by alterations of a broad lipid-related gene expression and increased H3K9 and H3K56 acetylation.**

(A) H&E staining and Oil red O staining of Sirt6 Floxed and Sirt6 LKO mice liver sections of 8 month old mice. Scale bars:100μm. (B) Expression levels of Sirt6 revealed by qPCR in Sirt6 Floxed or Sirt6 LKO mice. $n = 7$ BR; error bars = SEM. $t$-test. ****$p < 0.001$. (C) Expression levels of Sirt6 revealed by Western blot in Sirt6 Floxed or Sirt6 LKO mice. (D) RNA-seq showing downregulated different express genes enriched based on the biological process. $n = 3$ BR, $t$-test, $p$ value $< 0.05$, Log$_2$ FC $< -1$. (E) Metagene analysis showing the bind pattern of H3K9ac or H3K56ac in Sirt6 Floxed or Sirt6 LKO mice liver. (F) Venn diagram showing that RNA-sequence upregulated genes and ChIP-H3K9Ac/H3K56Ac peaks. (G) Heatmap showed the 47 candidate genes' ranking list.

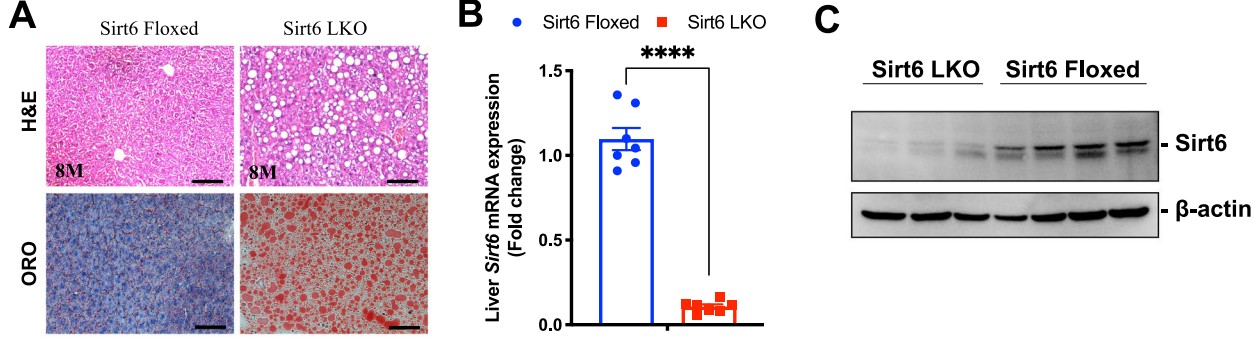

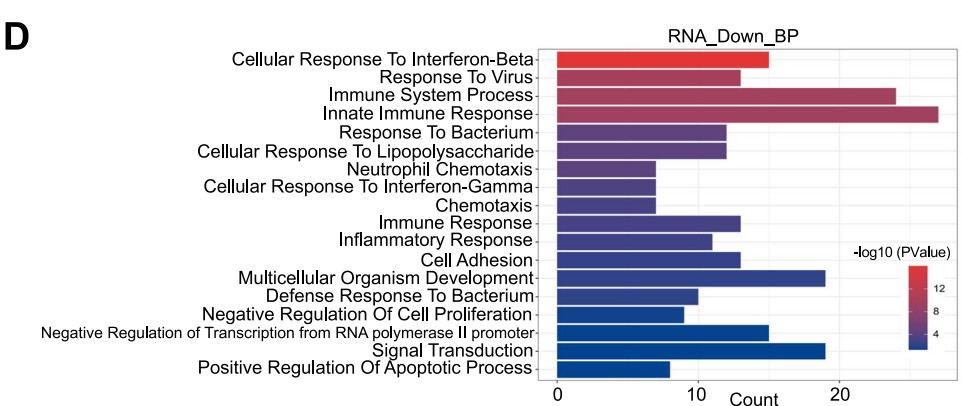

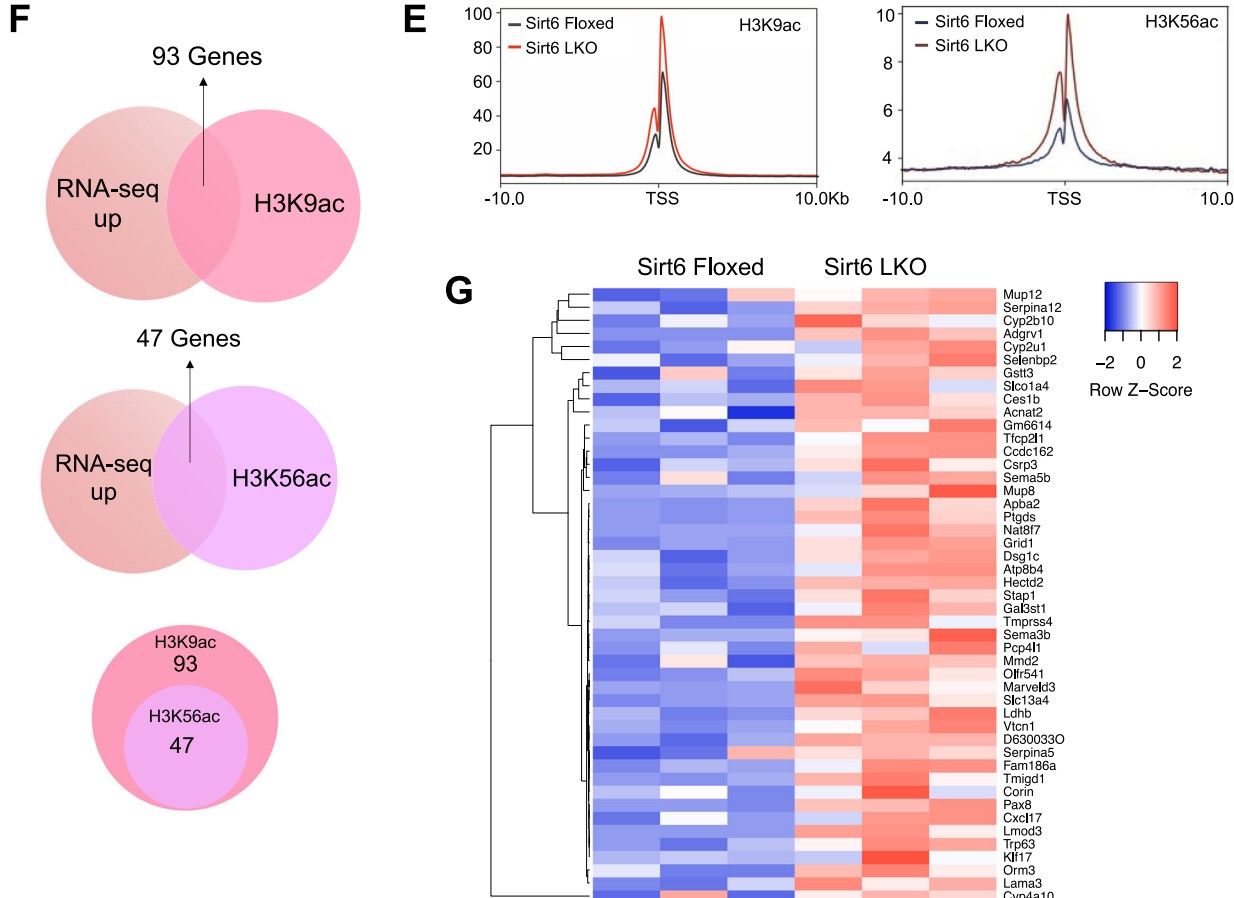

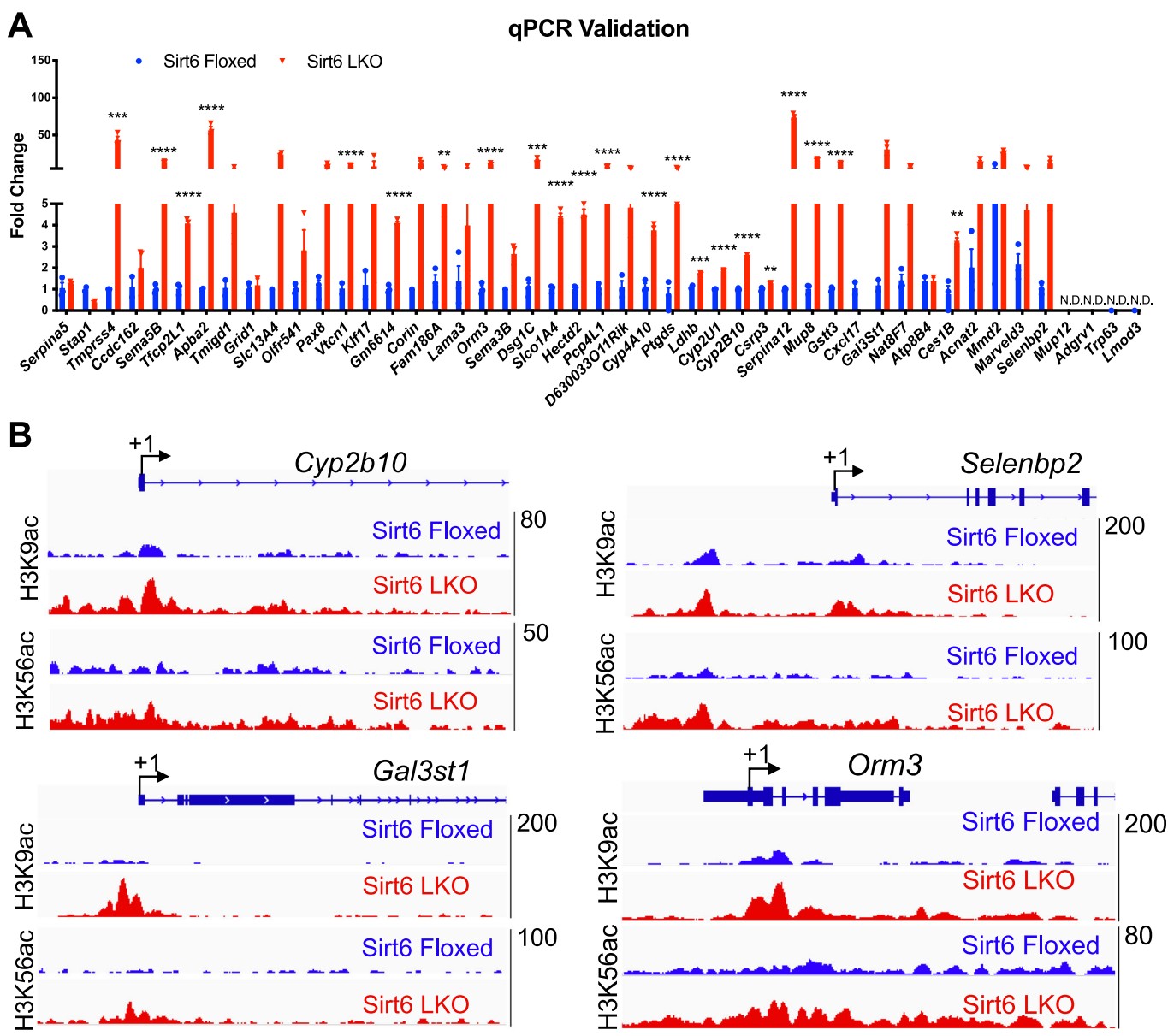

**Figure EV2. Validation of candidates' expression and the levels of H3K9Ac and H3K56Ac around their promoters.**

(A) qPCR validation of 47 candidate genes. Expression levels of Sirt6 revealed by qPCR in Sirt6 Floxed or Sirt6 LKO mice. $n = 3$ BR; error bars = SEM. Multiple $t$-test. $^{**}p < 0.01$, $^{***}p < 0.005$, $^{****}p < 0.001$. (B) IGV browser of images of read coverage across the 4 candidate genes binding peaks in H3K9Ac and H3K56ac ChIP-seq from Sirt6 Floxed versus Sirt6 LKO mice liver.

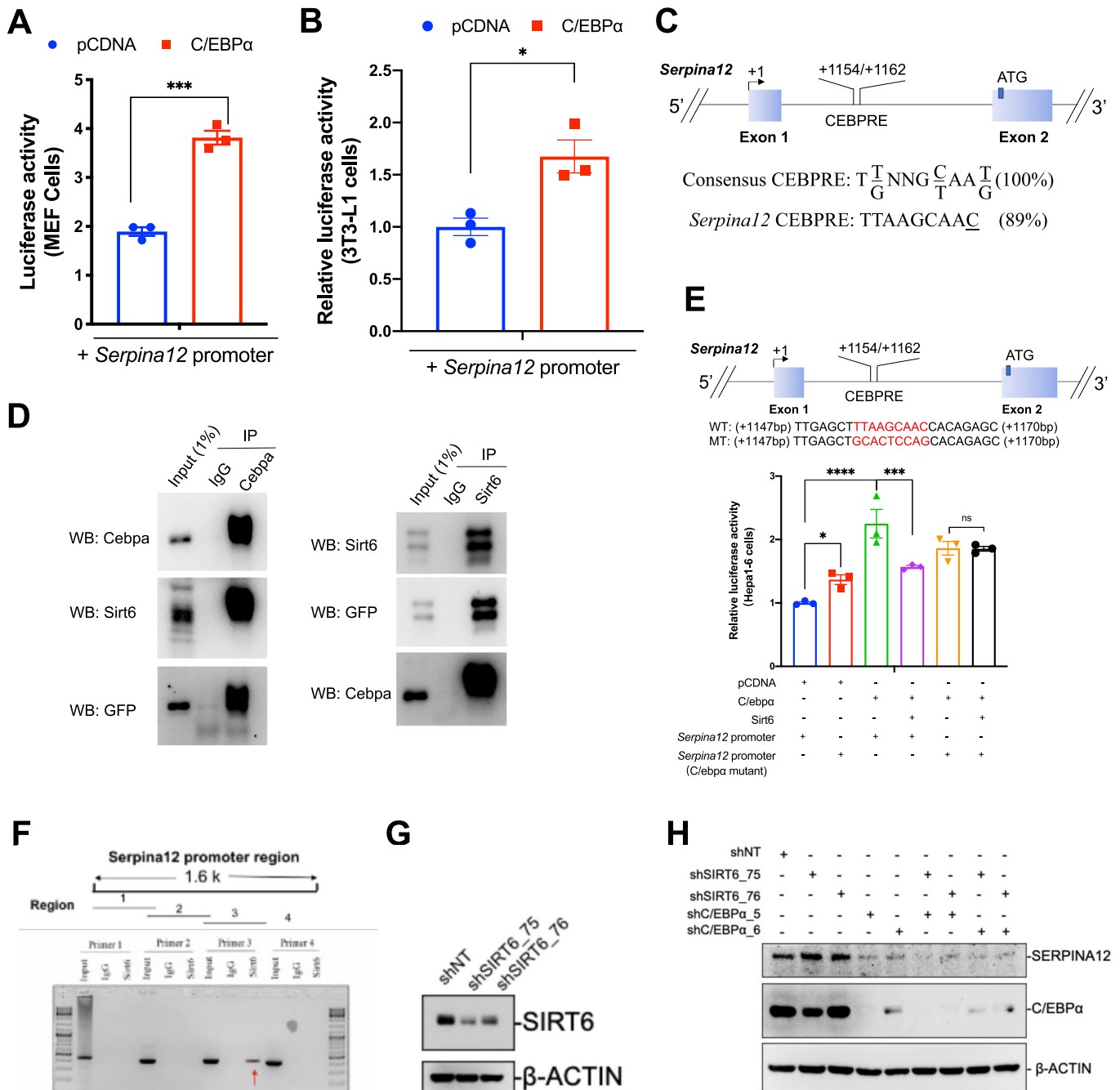

**Figure EV3. Serpina12 is upregulated in the liver after sirt6 ablation through transcription factor CEBPα.**

(A) Luciferase activity after transfected promoter *m-serpina12-0.75k* and CEBPα in MEF cells. *n* = 3 BR. (B) Luciferase activity after transfected promoter *m-serpina12-0.75k* and CEBPα in 3T3-L1 cells. *n* = 3 BR. (C) Consensus CEBPRE binding region on *Serpina12* gene region. (D) IP experiments on lysates from 293T/17 cells detected by antibodies of Sirt6 and C/ebpα. Sirt6-GFP and C/ebpα expression was induced 48 h prior to IP. (E) Ectopic expression of C/ebpα increased Serpina12 promoter activity, whereas mutation of C/ebpα-binding sites reduced the ienduction. Ectopic expression of Sirt6 represses the Serpina12 promoter activity but not the mutant C/ebpα-binding sites promoter activity. *n* = 3 BR. (F) SIRT6 ChIP to analysis SERPINA12 promoter binding region in HepG2 cells. (G) Western blot analysis shows the SIRT6 protein level in SIRT6 WT and knockdown groups in HepG2 cells. (H) Western blot analysis shows the SERPINA12 protein level with either knockdown SIRT6 or CEBPα in HepG2 cells. Data information: For (A),(B) and (E), error bars = SEM. *t*-test or two-way ANOVA. *$p < 0.05$, ***$p < 0.005$, ****$p < 0.001$, ns no significant difference.

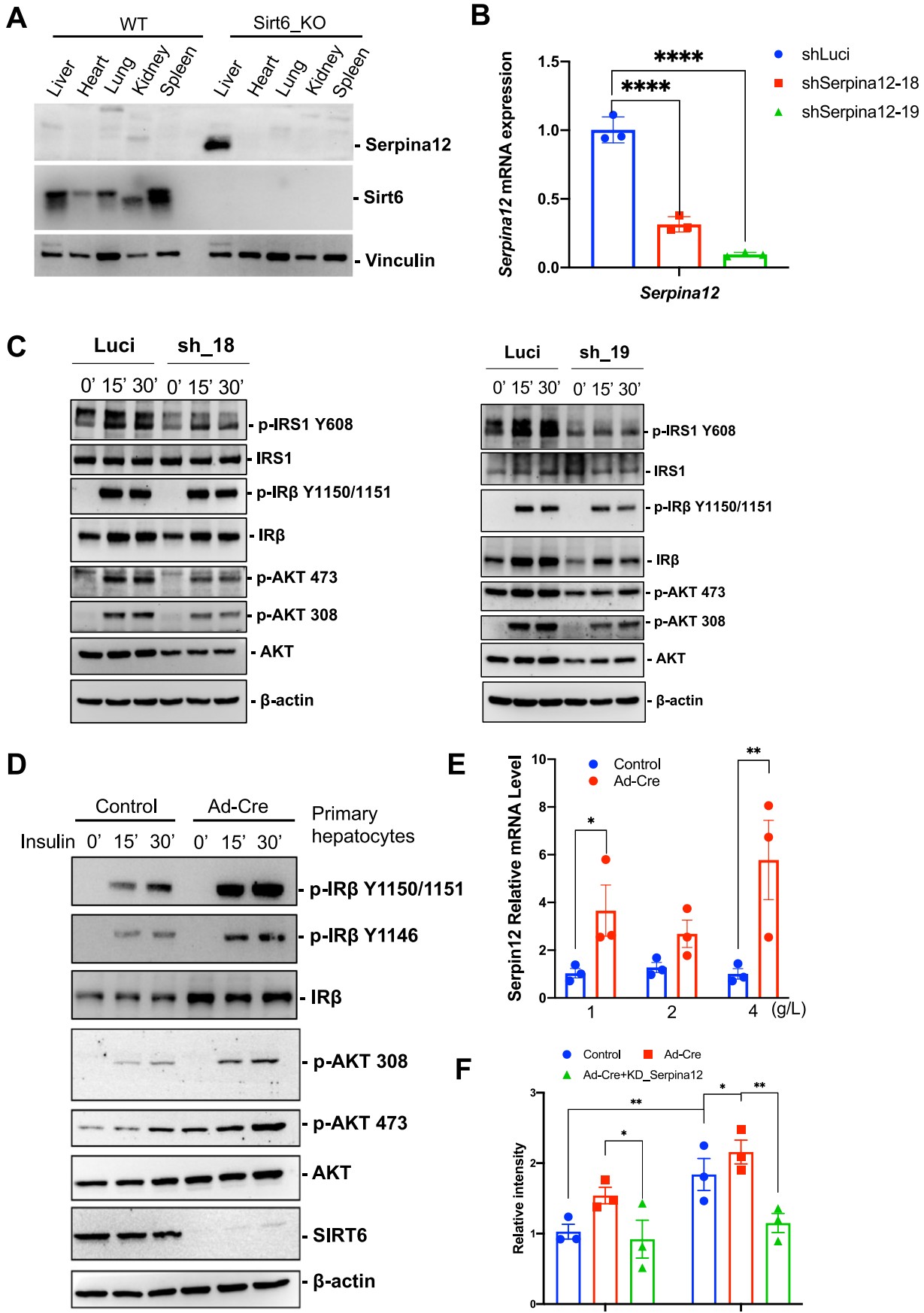

◀  **Figure EV4.   SIRT6 affects insulin signaling by regulating the expression of Serpina12.**

(A) Western blot shows the Serpina12 protein level in different tissue in SIRT6 WT or KO mice. (B) qPCR shows the knockdown efficiency of Serpina12 in primary hepatocytes. (C) Western blot analysis of insulin signaling following insulin treatment at 0 min, 15 min and 30 min by knockdown Luci control or two different Serpina12 shRNAs in primary hepatocytes. (D) Western blot analysis of insulin signaling following insulin treatment at 0 min, 15 min and 30 min in control or Ad-Cre in primary hepatocytes. (E) qPCR analysis of Serpina12 mRNA levels in different glucose concentration. (F) Relative intensity in control, knockout Sirt6 with or without knockdown Serpina12 primary hepatocytes. Data information: For (B),(E), and (F), $n = 3$ TR; error bars $=$ SEM. Multiple $t$-test or two-way ANOVA. *$p < 0.05$, **$p < 0.01$, ****$p < 0.001$.

                                                    

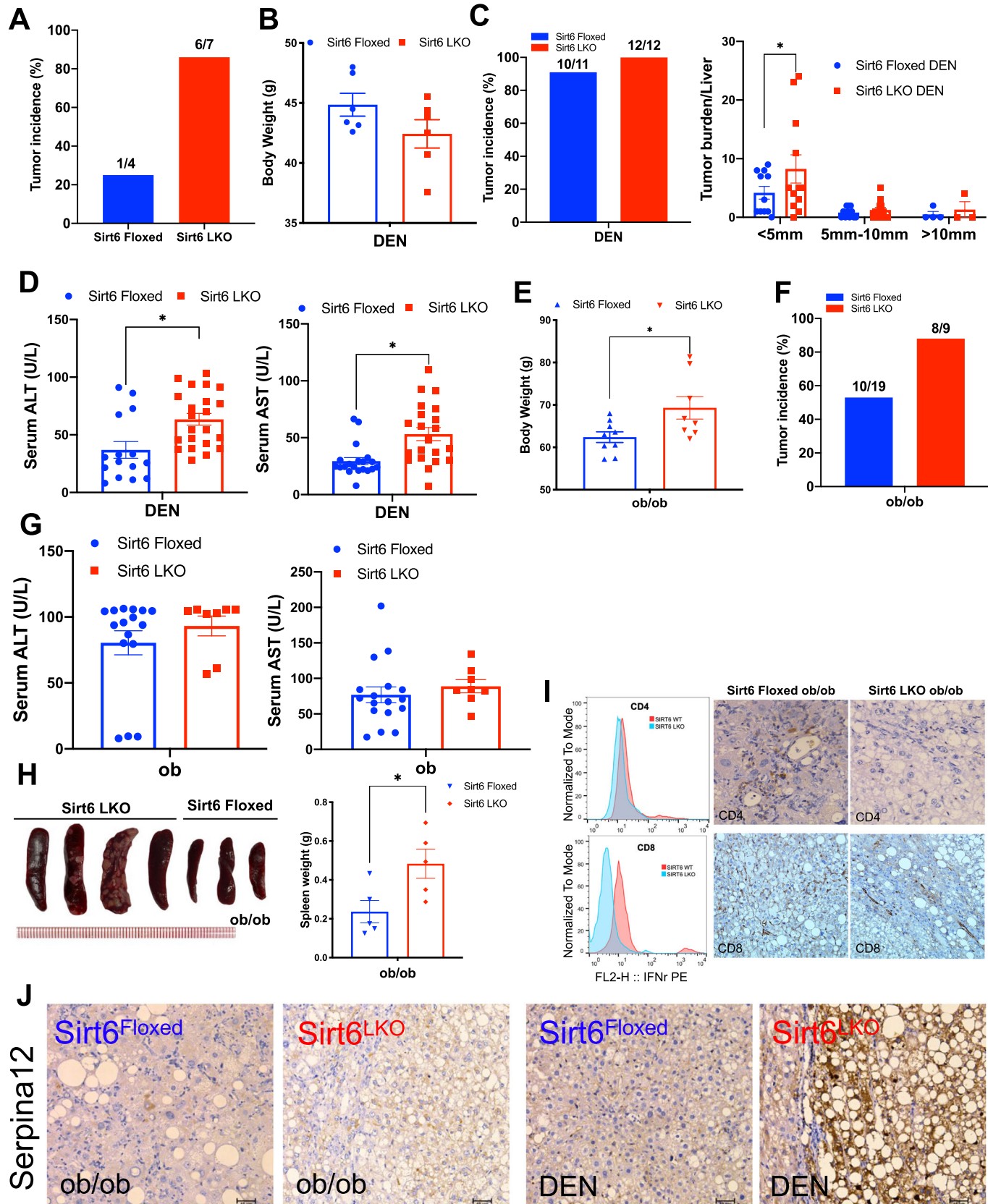

**Figure EV5. Sirt6 deficiency causes lipid-rich environment and less CD8⁺ T cells infiltration in the liver thus accelerate tumor formation.**

(A) Tumor incident (%) of Sirt6 Floxed or Sirt6 LKO mice in 2-year-old. (B) Body weight of Sirt6 Floxed and Sirt6 LKO with DEN injection mice. $n = 6$ BR. (C) Tumor incident in Sirt6 Floxed and Sirt6 LKO with DEN injection mice; Tumor burden per liver of Sirt6 Floxed and Sirt6 LKO with DEN injection mice. $n = 11$ or 12 BR. (D) Serum ALT level and AST level of Sirt6 Floxed and Sirt6 LKO with DEN injection mice. For ALT, $n = 15$ or 22 BR; For AST, $n = 20$ or 21 BR; (E) Body weight of Sirt6 Floxed and Sirt6 LKO with ob/ob mice. $n = 8$ or 9 BR. (F) Tumor incident in Sirt6 Floxed and Sirt6 LKO with ob/ob mice (6–9-month-old). (G) Serum ALT level and AST level of Sirt6 Floxed and Sirt6 LKO with ob/ob mice. For ALT, $n = 16$ or 8 BR; For AST, $n = 17$ or 8 BR; (H) Spleen morphology of Sirt6 Floxed and Sirt6 LKO with ob/ob mice at 7-month-old; Spleen weight of Sirt6 Floxed and Sirt6 LKO with ob/ob mice at 7-month-old. $n = 5$ BR. (I) The CD4⁺ and CD8⁺ T cells amount in the spleen of Sirt6 Floxed and Sirt6 LKO with ob/ob mice; IHC staining of CD4⁺ and CD8⁺ in Sirt6 Floxed and Sirt6 LKO with ob/ob mice. (J) IHC staining of Seprina12 in Sirt6 Floxed and Sirt6 LKO with ob/ob or DEN injection mice. Data information: For (B–E),(G), and (H), error bars = SEM. $t$-test. $*p < 0.05$.

   