## [Peer Review File · EMBO Reports]

Sirt6 ablation in the liver causes fatty liver that increases cancer risky by upregulating Serpina12

Licen Li, Jianming Zeng, Xin Zhang, Yangyang Feng, Josh Lei, Xiaoling Xu, Qiang Chen, and Chu-Xia Deng

Corresponding author(s): *Chu-Xia Deng (cxdeng@umac.mo)* , *Qiang Chen (qiangch@um.edu.mo)*

Review Timeline:

Submission Date:	20th Jul 23
Editorial Decision:	15th Aug 23
Revision Received:	13th Nov 23
Editorial Decision:	7th Dec 23
Revision Received:	11th Dec 23
Accepted:	9th Jan 24

Editor: *Deniz Senyilmaz Tiebe*

Transaction Report:

Dear Prof. Deng,

Thank you for the submission of your research manuscript to our journal, which was now seen by two referees, whose reports are copied below.

The referees express interest in the proposed role of Sirt6-Serpina12 link in HCC progression. However, they also raise significant concerns that need to be addressed to consider publication here.

While we agree with referee #2 that elucidating the mechanism by which Serpina12 regulates expression of genes related to lipogenesis would significantly strengthen the manuscript (point 3), not being able to do so will not preclude from publication here.

Given these positive recommendations, we would like to invite you to submit a revised manuscript. Please revise your manuscript with the understanding that the referee concerns (as in their reports) must be fully addressed and their suggestions taken on board. Please address all referee concerns in a complete point-by-point response. Acceptance of the manuscript will depend on a positive outcome of a second round of review. It is EMBO reports policy to allow a single round of major experimental revision only and acceptance or rejection of the manuscript will therefore depend on the completeness of your responses included in the next, final version of the manuscript.

We realize that it is difficult to revise to a specific deadline. In the interest of protecting the conceptual advance provided by the work, we recommend a revision within 3 months. Please discuss the revision progress ahead of this time with me if you require more time to complete the revisions, or if you have questions or comments regarding the revision (also by video chat).

1. A data availability section providing access to data deposited in public databases is missing (where applicable).
2. Your manuscript contains statistics and error bars based on $n=2$. Please use scatter plots in these cases.

You can submit the revision either as a Scientific Report or as a Research Article. For Scientific Reports, the revised manuscript can contain up to 5 main figures and 5 Expanded View figures, and it should not exceed 27000 characters. If the revision leads to a manuscript with more than 5 main figures it will be published as a Research Article. In this case the Results and Discussion section should be separate. If a Scientific Report is submitted, these sections have to be combined. This will help to shorten the manuscript text by eliminating some redundancy that is inevitable when discussing the same experiments twice. In either case, all materials and methods should be included in the main manuscript file.

3) We replaced Supplementary Information with Expanded View (EV) Figures and Tables that are collapsible/expandable online. A maximum of 5 EV Figures can be typeset. EV Figures should be cited as 'Figure EV1, Figure EV2' etc... in the text and their respective legends should be included in the main text after the legends of regular figures.

4) a .docx formatted letter INCLUDING the reviewers' reports and your detailed point-by-point responses to their comments. As part of the EMBO publication's Transparent Editorial Process, EMBO reports publishes online a Review Process File (RPF) to accompany accepted manuscripts. This File will be published in conjunction with your paper and will include the referee reports,

your point-by-point response and all pertinent correspondence relating to the manuscript.

<https://www.embopress.org/page/journal/14693178/authorguide#transparentprocess>

5) a complete author checklist, which you can download from our author guidelines

<https://www.embopress.org/page/journal/14693178/authorguide>. Please insert information in the checklist that is also reflected in the manuscript. The completed author checklist will also be part of the RPF.

6) Please note that all corresponding authors are required to supply an ORCID ID for their name upon submission of a revised manuscript (<<https://orcid.org/>>). Please find instructions on how to link your ORCID ID to your account in our manuscript tracking system in our Author guidelines

<<https://www.embopress.org/page/journal/14693178/authorguide#authorshipguidelines>>

7) Before submitting your revision, primary datasets produced in this study need to be deposited in an appropriate public database (see <https://www.embopress.org/page/journal/14693178/authorguide#datadeposition>). Please remember to provide a reviewer password if the datasets are not yet public. The accession numbers and database should be listed in a formal "Data Availability" section placed after Materials & Method (see also

<https://www.embopress.org/page/journal/14693178/authorguide#datadeposition>). Please note that the Data Availability Section is restricted to new primary data that are part of this study. * Note - All links should resolve to a page where the data can be accessed. *

Additional information on source data and instruction on how to label the files are available:

<https://www.embopress.org/page/journal/14693178/authorguide#sourcedata>

9) Our journal encourages inclusion of *data citations in the reference list* to directly cite datasets that were re-used and obtained from public databases. Data citations in the article text are distinct from normal bibliographical citations and should directly link to the database records from which the data can be accessed. In the main text, data citations are formatted as follows: "Data ref: Smith et al, 2001" or "Data ref: NCBI Sequence Read Archive PRJNA342805, 2017". In the Reference list, data citations must be labeled with "[DATASET]". A data reference must provide the database name, accession number/identifiers and a resolvable link to the landing page from which the data can be accessed at the end of the reference. Further instructions are available at <http://www.embopress.org/page/journal/14693178/authorguide#referencesformat>

10) Regarding data quantification (see Figure Legends:

<https://www.embopress.org/page/journal/14693178/authorguide#figureformat>)

11) The journal requires a statement specifying whether or not authors have competing interests (defined as all potential or actual interests that could be perceived to influence the presentation or interpretation of an article). In case of competing interests, this must be specified in your disclosure statement. Further information: <https://www.embopress.org/competing->

interests

12) Please also note our reference format:

I look forward to seeing a revised version of your manuscript when it is ready. Please let me know if you have questions or comments regarding the revision.

Kind regards,

Deniz Senyilmaz Tiebe

Deniz Senyilmaz Tiebe, PhD
Editor
EMBO Reports

Referee #1:

In this study, Deng and colleagues investigated the role of hepatic SIRT6 in regulation of HCC in vitro and in vivo.

Overall, this is a thorough and comprehensive study, containing well-designed in vivo phenotypic characterizations, solid in vitro mechanistic analyses and validations. The data is compelling, and all results are convincing. I only have a few suggestions/questions/comments for the improvement of this study:

1. The authors labeled their control mice as "WT" mice. But they are in fact *Sirt6* flox/flox mice based on my understanding. Suggest using the term of "Flox" instead of "WT" for these control mice.
2. One interesting observation is that SIRT6 LKO mice have reduced immune cells in the liver in both regular condition (Figure S1D) and after breeding into the ob/ob background (Figure S7I). The authors showed that there were reduced abundance of CD4+ and CD8+ T cells in Figure S7I. Have the authors checked Kupffer cells, the liver resident macrophages, in SIRT6 LKO mice under both conditions? Also, can the authors discuss the possible implications of this reduced immunity in promoting hepatic tumorigenesis in SIRT6 LKO mice?
3. Figure 3, very nice results! Two questions: does C/EBP α directly interact with SIRT6 so that SIRT6 can be specifically target to the promoter of *Serpina12* gene? Could the authors confirm that the binding of C/EBP α is important for SIRT6 to suppress the expression of *Serpina12* by generating mutant luciferase reporters that lack functional C/EBP α binding site (site 1 in Figure 3C), then show that SIRT6 repress the WT but not mutant luciferase reporter in hepatocytes?
4. Figure 4, does the SIRT6 level change (reduce) during high glucose or oleic acid induced lipogenesis in hepatocytes?
5. Figure 5, nice data. Have the authors analyzed the expression of SIRT6 in NASH? How about the correlations between SIRT6 and SERPINA12 as well as between SIRT6 and patient survival in HCC patients?

Referee #2:

In this study, Li et al demonstrated that depletion of *Sirt6* in mice liver resulted in alteration of H3K9 and H3K56 acetylation and gene expression profile. By comparison of the RNA-seq data and ChIP-seq data from wild type and *Sirt6*-LKO mice liver, *Serpina12* was identified as the key gene downstream of *Sirt6*. Ablation of *Sirt6* increased the expression level of *Serpina12* by enhancement of the level of H3K9ac and H3K56ac and the binding of transcription factor CEBP α . Increased expression of *Serpina12* in hepatocytes enhanced insulin signaling and lipid accumulation. In addition, *Sirt6* was shown to be a tumor suppressor in liver. Although the work might be interesting, several concerns are raised.

1. The authors demonstrated the depletion of *Sirt6* in hepatocyte resulted in high level of SERPINA12 to enhance insulin signaling and promote lipid accumulation. Knock out of *Sirt6* was shown to cause a lipid rich environment to accelerate hepatocellular carcinoma (HCC). Since insulin signaling was also reported to play important role in tumorigenesis and progression. The authors should evaluate the effect of insulin signaling in HCC.
2. SERPINA12 was reported to be an extracellular serine protease inhibitor which inhibited the activity of serine protease like KLK7 to increase the concentration of insulin in serum and enhance insulin signaling. In this study, *Serpina12* was shown to promote expression of genes related to glucose and lipid metabolism. Therefore, the author should demonstrate that SERPINA12 can be transported into nucleus.
3. An interesting question should be addressed is the mechanism of SERPINA12 be recruited to chromatin. Either the DNA motif or the protein factors targeting SERPINA12 to chromatin should be identified.

Referee #1:

In this study, Deng and colleagues investigated the role of hepatic SIRT6 in regulation of HCC in vitro and in vivo.

Overall, this is a thorough and comprehensive study, containing well-designed in vivo phenotypic characterizations, solid in vitro mechanistic analyses and validations. The data is compelling, and all results are convincing. I only have a few suggestions/questions/comments for the improvement of this study:

1. The authors labeled their control mice as "WT" mice. But they are in fact *Sirt6* flox/flox mice based on my understanding. Suggest using the term of "Floxed" instead of "WT" for these control mice.

Thank you for your suggestion. We have changed the term "WT" to "Floxed" accordingly in the revised manuscript.

2. One interesting observation is that SIRT6 LKO mice have reduced immune cells in the liver in both regular condition (Figure S1D) and after breeding into the *ob/ob* background (Figure S7I). The authors showed that there were reduced abundance of CD4+ and CD8+ T cells in Figure S7I. Have the authors checked Kupffer cells, the liver resident macrophages, in SIRT6 LKO mice under both conditions? Also, can the authors discuss the possible implications of this reduced immunity in promoting hepatic tumorigenesis in SIRT6 LKO mice?

For detecting Kupffer cells, we did IHC with an antibody to IBA1, which is Kupffer cells marker in the liver in *Sirt6* LKO mice in both conditions. There were no obvious changes in both conditions (R-Figure 1A and 1B).

R-Figure 1 (A) IHC staining of IBA1 in *Sirt6* Floxed and *Sirt6* LKO mice. (B) IBA1 IHC staining quantification. (C) IHC staining of IBA1 in *Sirt6* Floxed and *Sirt6* LKO mice under the condition of obesity. (D) IBA1 IHC staining quantification.

We observed that the immune response related pathway was decreased in *Sirt6* LKO mice using RNA-seq

analysis. These pathways were downregulated, suggesting the weaker activity of the immune cells in the Sirt6 LKO mice liver. Although we found that the number of Kupffer cells has no obvious change in the liver, T cells could also affect immune processes. In Sirt6 LKO mice with obesity background, we have found reduced CD4+ and CD8+ T cells in the liver. The reduced immunity could lead to the failure to eliminate cancer cells in the liver, thus promoting the tumor formation. However, the relationship between the immune system and cancer is complex, and enhancing the immune system alone may not be effective to treat liver cancer in clinical condition. Further study is needed to illustrate the effect in this case. The possible implications were discussed in the revised manuscript. We have indicated in the Discussion that the reduced CD4+ and CD8+ T cells in the liver could lead to the failure to eliminate cancer cells in the liver, thus promoting the tumor formation (page 28, line 1-2, line 11-19 and page 29 line 1-6).

3. Figure 3, very nice results! Two questions: does C/EBPa directly interact with SIRT6 so that SIRT6 can be specifically target to the promoter of Serpina12 gene? Could the authors confirm that the binding of C/EBPa is important for SIRT6 to suppress the expression of Serpina12 by generating mutant luciferase reporters that lack functional C/EBPa binding site (site 1 in Figure 3C), then show that SIRT6 repress the WT but not mutant luciferase reporter in hepatocytes?

To determine whether C/EBPa could directly interact with SIRT6, we induced Sirt6-GFP and C/ebpa expression 48h in 293T cells and performed a co-immunoprecipitation assay with antibodies against Sirt6 and C/ebpa. The results showed that CEBPA could directly interact with SIRT6 (R-Figure 2A).

To address the second question, we generated a mutant luciferase reporter that lacks functional C/EBPa binding site. The data indicated that Sirt6 could suppress the increased expression of WT Serpina12-luc activity induced by C/EBPa. In contrast, such suppression was largely abolished after the mutation of the binding site of C/EBPa (R-Figure 2B). This observation indicates that the binding of C/EBPa is important for SIRT6 to suppress the expression of Serpina12. This data was placed in the Figure Extended View 3D and 3E and the corresponding text (page 13 and line 6-11).

R-Figure 2 (A) IP experiments on lysates from 293T/17 cells detected by antibodies of Sirt6 and C/ebpa. Sirt6-GFP and C/ebpa expression was induced 48 h prior to IP. (B) Ectopic expression of C/ebpa increased Serpina12 promoter activity, whereas mutation of C/ebpa-binding sites reduced the activity. Ectopic expression of Sirt6 represses the Serpina12 promoter activity but not the mutant C/ebpa-binding sites promoter activity.

4. Figure 4, does the SIRT6 level change (reduce) during high glucose or oleic acid induced lipogenesis in hepatocytes?

Thank you for your comment. We have observed that Sirt6 level has no significant change in 1g/L (low glucose) compared to 4g/L (high glucose) both at RNA (R-Figure 3A) and protein levels (manuscript Figure 4D). However, the SIRT6 level has significantly decreased after oleic acid treatment in HepG2 cells (R-Figure 3B and 3C), suggesting that the expression change of Sirt6 might be more sensitive to condition under lipogenesis.

R-Figure 3 (A) qPCR analysis of Sirt6 mRNA levels in different glucose concentration. (B) qPCR analysis of Sirt6 mRNA levels in different oleic acid concentration treatment. (C) Western blot analysis of Sirt6 mRNA levels in different oleic acid concentration treatment.

5. Figure 5, nice data. Have the authors analyzed the expression of SIRT6 in NASH? How about the correlations between SIRT6 and SERPINA12 as well as between SIRT6 and patient survival in HCC patients?

Thank you for the suggestion. We checked the expression of SIRT6 in NASH patients in a GEO database and found that it was at a low expression level compared to other genes (revised manuscript Figure 5D).

We found a negative correlation between Sirt6 and Serpina12 in the patients with HCC in a GSE25097 database (R-Figure 4A). To understand the role of SIRT6 for patient survival, we analyzed the TCGA database in HCC patients. We defined SIRT6 high or SIRT6 low expression based on the mean value in these HCC patients and the data indicated that overall survival has no significant difference between SIRT6 high and SIRT6 low expression (R-Figure 4B).

R-Figure 4 (A) Pearson correlation analysis of SIRT6 and SERPINA12 mRNA expression in HCC (GSE25097).

(B) Kaplan-Meier curves for overall survival of HCC patients with high and low SIRT6.

Referee #2:

In this study, Li et al demonstrated that depletion of Sirt6 in mice liver resulted in alteration of H3K9 and H3K56

acetylation and gene expression profile. By comparison of the RNA-seq data and ChIP-seq data from wild type and Sirt6-LKO mice live, Serpina12 was identified as the key gene downstream of Sirt6. Ablation of Sirt6 increased the expression level of Serpina12 by enhancement of the level of H3K9ac and H3K56ac and the binding of transcription factor CEBP α . Increased expression of Serpina12 in hepatocytes enhanced insulin signaling and lipid accumulation. In addition, Sirt6 was shown to be a tumor suppressor in liver. Although the work might be interested, several concerns are raised.

1. The authors demonstrated the depletion of Sirt6 in hepatocyte resulted in high level of SERPINA12 to enhance insulin signaling and promote lipid accumulation. Knock out of Sirt6 was shown to cause a lipid rich environment to accelerate hepatocellular carcinoma (HCC). Since insulin signaling was also reported to play important role in tumorigenesis and progression. The authors should evaluate the effect of insulin signaling in HCC.

Thank you for the suggestion. Insulin-dependent signaling pathways are often overactivated in human HCC due to the overexpression of signaling components and loss of negative regulators^[1, 2]. As a result, hyperinsulinemia can directly promote cancer cell metabolism, proliferation, and survival, potentially contributing to HCC development and progression^[3]. We have tested the effect of insulin signaling in HCC and the data indicated that insulin signaling plays a crucial role in the development of hepatocellular carcinoma (HCC) by promoting cell proliferation (R-Figure 5). The data is placed in the revised manuscript Appendix Figure S3, and the corresponding text (page 26, line 6-12).

R-Figure 5 Cell proliferation rate with different dose of insulin treatment.

2. SERPINA12 was reported to be an extracellular serine protease inhibitor which inhibited the activity of serine protease like KLK7 to increase the concentration of insulin in serum and enhance insulin signaling. In this study, Serpina12 was shown to promote expression of genes related to glucose and lipid metabolism. Therefore, the author should demonstrate that SERPINA12 can be transported into nucleus.

Thank you for the suggestion. To understand the location of Serpina12, we isolate cytoplasm extract and nuclear extract in primary hepatocytes. The data that Serpina12 was in cytoplasm in the cells, but not in the nucleus (R-Figure 6). We have also demonstrated that knockdown Sirt6 increased Serpina12 expression in the cytoplasm, which may trigger the lipid accumulated in the cells. Thus, effect of Serpina12 on genes related to glucose and lipid metabolism might be caused other mechanism rather than directly regulates their expression.

R-Figure 6 Western blot analysis of Serpina12 in cytoplasm extract or nuclear extract with or without knockout Sirt6 in primary hepatocytes.

3. An interesting question should be addressed is the mechanism of SERPINA12 be recruited to chromatin. Either the DNA motif or the protein factors targeting SERPINA12 to chromatin should be identified.

In our study, we found that Sirt6 affects the Histone 3 lysine 9 acetylation, and they have increased binding peaks in the Serpina12 locus. As Serpina12 is only in the cytoplasm, but not in the nucleus (R-Figure 6), it is unlikely that Serpina12 could be recruited to chromatin.

- [1] Alqahtani, A., Z. Khan, A. Alloghbi, et al., Hepatocellular Carcinoma: Molecular Mechanisms and Targeted Therapies[J]. *Medicina (Kaunas)*, 2019. **55**(9).
- [2] Enguita-German, M., P. Fortes, Targeting the insulin-like growth factor pathway in hepatocellular carcinoma[J]. *World J Hepatol*, 2014. **6**(10): 716-37.
- [3] Chettouh, H., M. Lequoy, L. Fartoux, et al., Hyperinsulinaemia and insulin signalling in the pathogenesis and the clinical course of hepatocellular carcinoma[J]. *Liver Int*, 2015. **35**(10): 2203-17.

Dear Prof. Deng,

Thank you for submitting your revised manuscript. It has now been seen by both of the original referees.

As you can see, the referees find that the study is significantly improved during revision and recommend publication. However, I need you to address the points below before I can accept the manuscript.

- Please remove the figures from the main manuscript file and reorder the sections as follows: Title page - Abstract & Keywords - Introduction - Results - Discussion - Materials & Methods - Data Availability - Acknowledgments - Disclosure Statement & Competing Interests - References - Figure Legends - Tables with legends - Expanded View Figure Legends.
- Please remove the Author Contributions section from the manuscript.
- We note that the ORCID iDs of Dr. Deng and Dr. Chen are currently missing. EMBO Press policy asks for all corresponding authors to link to their ORCID iDs. Please see "Authorship Guidelines" in the Guide to Authors here: <https://www.embopress.org/page/journal/14693178/authorguide#authorshipguidelines>

In order to link your ORCID iD to your account in our manuscript tracking system, please do the following:

1. Click the 'Modify Profile' link at the bottom of your homepage in our system.
2. On the next page you will see a box halfway down the page titled ORCID*. Below this box is red text reading 'To Register/Link to ORCID, click here'. Please follow that link: you will be taken to ORCID where you can log in to your account (or create an account if you don't have one)
3. You will then be asked to authorise Wiley to access your ORCID information. Once you have approved the linking, you will be brought back to our manuscript system.

We regret that we cannot do this linking on your behalf for security reasons.

- Please make sure that the funding information is included in the Acknowledgements section, as well.
- We note that you submitted three tables listing the reagents used (RT primers, Antibody list, sgRNA sequence). I encourage converting these tables into the format of structured methods (please see examples and templates here: <https://www.embopress.org/page/journal/14693178/authorguide#structuredmethods>)
- Please add more details about the experiments performed, statistical tests applied and the meaning of *, ** and *** to the legend of Appendix Figure S3.
- Please fill in the Source Data checklist (attached).
- Please resubmit Source Data as one zip file per figure.
- Abbreviations section needs to be removed from the manuscript. Abbreviations should be defined in brackets after their first mention in the text, not in a list of abbreviations.
- Please make a note in the legends of the following figures regarding the reuse of the panels of
 - o Figure 1A and Figure EV1A
 - o Figure 1A and Figure 7
 - o Figure 6C and Figure 7
 - o Figure 7 and Figure EV1A.
- Please make the bioproject datasets listed in the Data Availability section publicly available and remove the reviewer access links from the manuscript. Instead, please add links that directly resolve to the datasets.
- Our production/data editors have asked you to clarify several points in the figure legends:
 1. Please indicate the statistical test used for data analysis in the legends of figures 1b, c, d, f; EV1d
 2. Please note that for the figure 4f, p-values and statistical tests are indicated in the legends. However, comparison for the same, ""**/"" has not been represented in the figures. Please rectify this in the figures or legends as applicable.
 3. Please note that information related to n is missing in the legend of figure 1b.
 4. Although 'n' is provided, please describe the nature of entity for 'n' in the legends of figures 2c, d, e, g; 3b, d-h; 4f; 5a, c; EV1b; EV5b-e, g, h."

Thank you again for giving us to consider your manuscript for EMBO Reports, I look forward to your minor revision.

Kind regards,

Deniz Senyilmaz Tiebe

--

Deniz Senyilmaz Tiebe, PhD
Editor
EMBO Reports

Referee #1:

The revision addressed my major concerns.

Referee #2:

The authors addressed my previous concerns. I have no question now.

All editorial and formatting issues were resolved by the authors.

Dear Prof. Deng,

Thank you for submitting your revised manuscript. My apologies for the unusual delay in getting back to you. I have now looked at everything and all is fine. Therefore, I am very pleased to accept your manuscript for publication in EMBO Reports.

Congratulations on a nice work!

Kind regards,

Deniz Senyilmaz Tiebe

--

Deniz Senyilmaz Tiebe, PhD
Scientific Editor
EMBO Reports

--
